# Investigating the Spatial Heterogeneity of Urban Heat Island Responses to Climate Change Based on Local Climate Zones

**Fei He** [1]**, Luyun Liu** [1,]*****, Yu Huang** [2,]*****, Komi Bernard Bedra** [3] **and Minhuan Zhang** [1,4]

1    School of Landscape Architecture, Central South University of Forestry and Technology,
     Changsha 410004, China
2    School of Art & Design, Nanning University, Nanning 530200, China
3    School of Architecture and Art, Central South University, Changsha 410083, China
4    Hunan Big Data Engineering Technology Research Center of Natural Protected Areas Landscape Resource,
     Changsha 410004, China
*    Correspondence: t20172369@csuft.edu.cn (L.L.); huangyu@nnxy.edu.cn (Y.H.); Tel.: +86-18684941310 (L.L.)

**Abstract:** Global warming and the urban heat island (UHI) phenomenon have significant impacts on human activities, against which it is necessary to develop effective coping strategies. Based on the local climate zone (LCZ) system, this study used the land-cover and surface temperature data on the Chang–Zhu–Tan (CZT) urban agglomeration in China in 2006, 2010, 2016, and 2020 to analyze the impact of climate change on the land surface temperature (LST) under different land-cover types. The results illustrate that the LCZ map generated on the basis of the improved World Urban Database and Access Portal Tools (WUDAPT) is more accurate and efficient than the traditional method. The accuracy is increased by more than 15%. From 2006 to 2020, the main built-up types in the CZT urban agglomeration were the sparsely built, the large low-rise, and the compact mid-rise types. The low-plant type represents the most significant proportion of the natural types, followed by the water and the dense-tree types. The built-up types in the CZT urban agglomeration tend to be the high-rise, dense, and industrial types. Urban construction land is taken mainly from the sparsely built type of land. The average LST of the large low-rise and heavy-industry zones is significantly higher than the average LST of the three cities. The average LST values for the water and dense-tree zones are significantly lower than the other average LST values. The LST is stable in each LCZ, showing little correlation with the size of the LCZ area. Compact low-rise land use is ineffective against climate warming and inhibits economic growth. Compact high-rise and open high-rise land can not only effectively deal with climate warming but can also significantly stimulate economic growth. This paper helps us to understand the effect of land cover on climate warming and the economic benefits of LCZs in the CZT urban agglomeration and provides strategies to optimize the use of land resources.

**Keywords:** urban heat island; spatial heterogeneity; land surface temperature; economic effect; local climate zone; remote sensing; WUDAPT; urban agglomeration

## 1. Introduction

Global air and land temperatures have shown a long-term warming trend since preindustrial times [1]. Regions with warmer temperatures suffer from stronger urban heat islands (UHIs) [2]. UHI, a phenomenon in which a region's air temperature or land surface temperature (LST) is typically greater than that of its surrounding areas [3], is directly linked to anthropomorphic activities, such as the direct heat emissions of fuel combustion and electricity consumption [4,5]. UHI affects people's lives and the environment in many ways, which not only affects the quality of human life but also affects the local climate, seriously hindering the sustainable development of the urban environment [6–8]. The deterioration of the environment limits the social and economic benefits of land use [9]. A local climate zone (LCZ) classifies the land surface according to different building densities, geometric shapes, and surface characteristics and can intuitively describe land development. Therefore, it is

helpful for the rational development of urban land to clarify the relationship between the UHI phenomenon and LCZ changes and determine their economic implications to explore the factors that influence them.

Researchers initially measured the UHI phenomenon via the dichotomy between urban and rural areas [10]. These two conceptions, however, are vague. For this reason, in 2012, Stewart and Oke proposed an LCZ classification system [6,11]. LCZ classification provides a clearer research framework for studies on the UHI phenomenon by dividing land cover into 17 types. In recent years, researchers have studied UHI on the basis of using LCZ classification in many cities around the world [12]. Mu, for instance, studied the spatial distribution of the LST on the basis of LCZ [13]. Mushore and Zhou studied the influence of the evolution of LCZ on the LST [14–16]. Pozo and Wang studied the seasonal variation in the LST for different LCZ types [17,18]. Cai and Geng compared the changes in the LST with different LCZ structures [19,20]. Geletič, Wang, and Lu studied the impact of the spatiotemporal evolution of LCZ on the LST in different cities [21–23]. Zhao studied the variation in the LST with different climatic backgrounds and LCZs [24]. Li distinguished the cooling and heating effects of LCZ [25]. Researchers have found that land cover, solar radiation, surface albedo, urban form, human activities, and climate change can all have significant effects on LST [22,25–32]. Studies have shown that UHI is strongest in the summer [25]. The increase in the daytime LST is more pronounced than the increase in the nighttime LST [22]. A high surface albedo reduces the conversion of daylight into heat [29,30]. The LST of urban built-up land is higher than that of natural surfaces [22]. The natural surface has a very significant cooling effect [25]. Areas with dense buildings have a higher LST than areas with sparse buildings [14]. Lu found that the evolution of LCZ has been affected by the level of economic development and local policies [23]. Studies have proposed that in order to alleviate the UHI phenomenon, urban planning should rationally arrange parks, green spaces, and water ponds; avoid high-density, low-rise residential areas with low vegetation coverage; increase inner-city surface roughness; and use reflective roofs, green roofs, and vertical greening systems [29,30,33–35].

In the process of studying the UHI phenomenon, identifying its scope is a fundamental issue. On the other hand, it is also necessary to quantitatively characterize the urban heat island intensity. At present, the urban and rural temperature thresholds, the temperature grade threshold, the Gaussian fitting parameter, and the temperature decay mutation are used mainly to identify the UHI scope. The urban and rural temperature difference, the temperature grade difference, the Gaussian surface height, and the temperature attenuation amplitude methods are used mainly to quantify the urban heat island intensity [25,36–45]. Remote-sensing inversion or field measurements are usually used to obtain LST data [21,26]. Pearson's correlation coefficient is commonly used to study the strength of the linear correlation between two variables—in this case, LCZ and the LST [7,23,46].

In 2015, the World Urban Database and Access Portal Tools (WUDAPT) developed an LCZ generator to provide researchers around the world with a way to generate LCZ maps. In 2021, the generator simplified the workflow, requiring only a valid training area file and some metadata as inputs. The web application automatically generates LCZ maps (available from https://LCZ-generator.rub.de, accessed on 15 September 2022) [47]. The training areas are usually manually obtained through Google Earth [19,22]. The larger the training areas per LCZ class, the better the classification [47]. The workload will be enormous when using the traditional WUDAPT method to obtain training areas for multiple objects. In light of this problem, Lu utilized the time series sample migration method to more efficiently obtain the training areas of a city for multiple years [23,48]. To improve the efficiency of obtaining LCZ maps of multiple cities simultaneously, Cai first classifies a large area and then generates an LCZ map of the large area before extracting LCZ maps of each city in the area [19].

At present, there are relatively few studies on the spatial heterogeneity of UHI changes along with weather temperatures. Moreover, we must efficiently generate LCZ maps for multiple years in adjacent cities. In view of this, we propose an improved WUDAPT

method and take the LCZ system as the framework to analyze the following: (1) the correlation between LST and the total areas of LCZ; (2) which LCZ types are most affected by high weather temperatures; and (3) which industries' GDP figures are affected the most by LCZ changes. The purpose of this is to judge the following: (1) whether the change in the total area of an LCZ type will affect the average LST; (2) how the LST in different LCZ types will vary according to weather temperature changes; and (3) the economic benefits of each LCZ. Finally, we provide relevant recommendations for urban construction. The innovation of this paper is as follows: First, we use an improved WUDAPT sample generation method to generate LCZ maps of multiple cities at multiple times. Second, we investigate the spatial heterogeneity of UHI responses to weather temperature changes by looking at the LCZ. Third, this paper explores the relationship between LCZ changes and urban economic growth.

## 2. Method

### 2.1. Study Area

The Chang–Zhu–Tan (CZT) urban agglomeration is in Hunan Province, China (Figure 1), and includes three cities: Changsha, Zhuzhou, and Xiangtan. It is a significant urban agglomeration in the middle reaches of the Yangtze River and the main growth pole of Hunan Province's economic development. In 2007, the state approved the CZT urban agglomeration as a pilot area for comprehensive social construction reform. The aim is to build them into "two-type" cities (national resource-saving type and national environmentally friendly type) [49]. The Xiangjiang River flows from south to north through the three cities. The CZT urban agglomeration has a subtropical monsoon climate with hotter thermal conditions than other cities in Hunan Province. In China, Changsha is one of the four hottest cities during the summer. The high temperatures in Hunan Province are concentrated in July and August. In the past 5 years, the average July temperature in Hunan Province has risen by 0.3 °C, from 29.2 °C to 29.5 °C. In Changsha, Zhuzhou, and Xiangtan, the average temperature in July increased by 0.1 °C, 1.0 °C, and 0.6 °C over the same period, respectively. In 2022, there were 85 days when the average daily temperature in summer was above 30 °C and approximately 30 days when the temperature was higher than 35 °C. This shows a serious urban heat island problem in the CZT urban agglomeration [50–52]. The UHI of the CZT urban agglomeration is distributed mainly in the built-up area with growing intensity. In the past 2 decades, the UHI in the central urban area of the CZT urban agglomeration has gradually increased, from 3.3 °C to 10.1 °C. This paper selects the built-up area of the CZT urban agglomeration as the study area. The study areas are Changsha (28°10′36″ N, 112°58′39″ E, 1008.32 km$^2$), Zhuzhou (27°51′13″ N, 113°07′55″ E, 281.19 km$^2$), and Xiangtan (27°51′32″ N, 112°55′51″ E, 309.21 km$^2$).

### 2.2. Materials and Methods

The research framework in this paper is divided into three steps (Figure 2). First, we generate local climate zone (LCZ) maps by using a modified WUDAPT approach. The land surface temperature (LST) maps are generated by using Envi5.3 software with the remote-sensing data of the CZT urban agglomeration in 2006, 2010, 2016, and 2020. Next, the accuracy of the maps is verified. Second, we extract the area and LST of each LCZ in the three cities and make the transfer matrix of LCZs for each city by using the spatial analyst and data management tools in ArcGIS. We classify LST by using the mean standard deviation method. Third, we apply the Pearson correlation analysis model to combine the urban GDP data of the three cities, the LCZ area data, and the LST data in 2006, 2010, 2016, and 2020. We build three models: (1) the LCZ area and their corresponding LST; (2) the weather temperature and the LST difference; and (3) the LCZ area change rate and the GDP change rate.

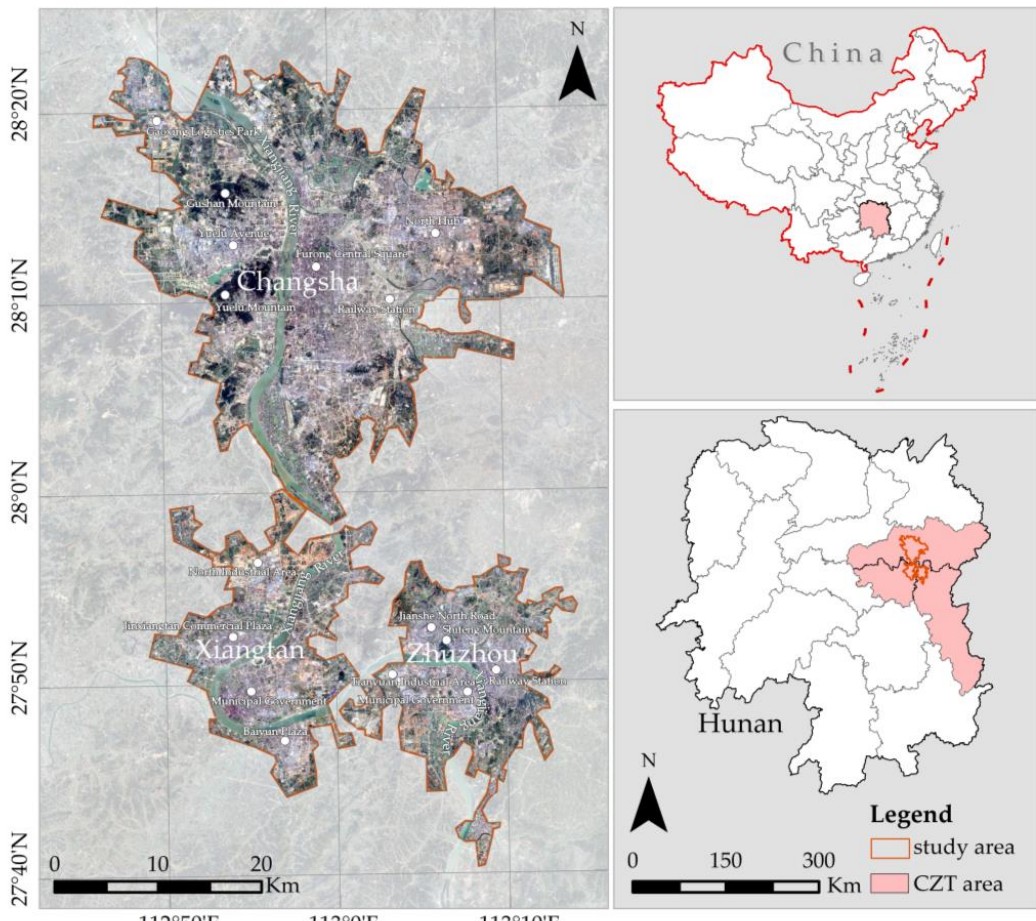

**Figure 1.** Location of the study areas.

### 2.2.1. Generation of the LCZ Map

1. LCZ system

Local climate zones (LCZs) are defined as regions of uniform surface cover, structure, material, and human activity that span hundreds of meters to several kilometers [22–25]. The LCZ system divides land cover into 10 built-up types and seven natural types according to the characteristics of the area. Built-up types are divided into compact high-rise (LCZ 1), compact mid-rise (LCZ 2), compact low-rise (LCZ 3), open high-rise (LCZ 4), open mid-rise (LCZ 5), open low-rise (LCZ 6), lightweight low-rise (LCZ 7), large low-rise (LCZ 8), sparsely built (LCZ 9), and heavy-industry (LCZ 10) types. The natural types are divided into dense-tree (LCZ A), scattered tree (LCZ B), bush and scrub (LCZ C), low-plant (LCZ D), bare-rock or paved (LCZ E), bare-soil or sand (LCZ F), and water (LCZ G) types [6].

The Chang–Zhu–Tan (CZT) urban agglomeration has few lightweight low-rise buildings, scattered trees, bushes, scrub, and large areas of bare rock or pavement. According to the master plan of the CZT urban agglomeration [49], the urban exposed construction land comprises mainly large low-rise buildings, so the exposed construction land is classified as large low-rise. In this paper, 12 types of LCZs were identified in the urban area, as is shown in Table 1.

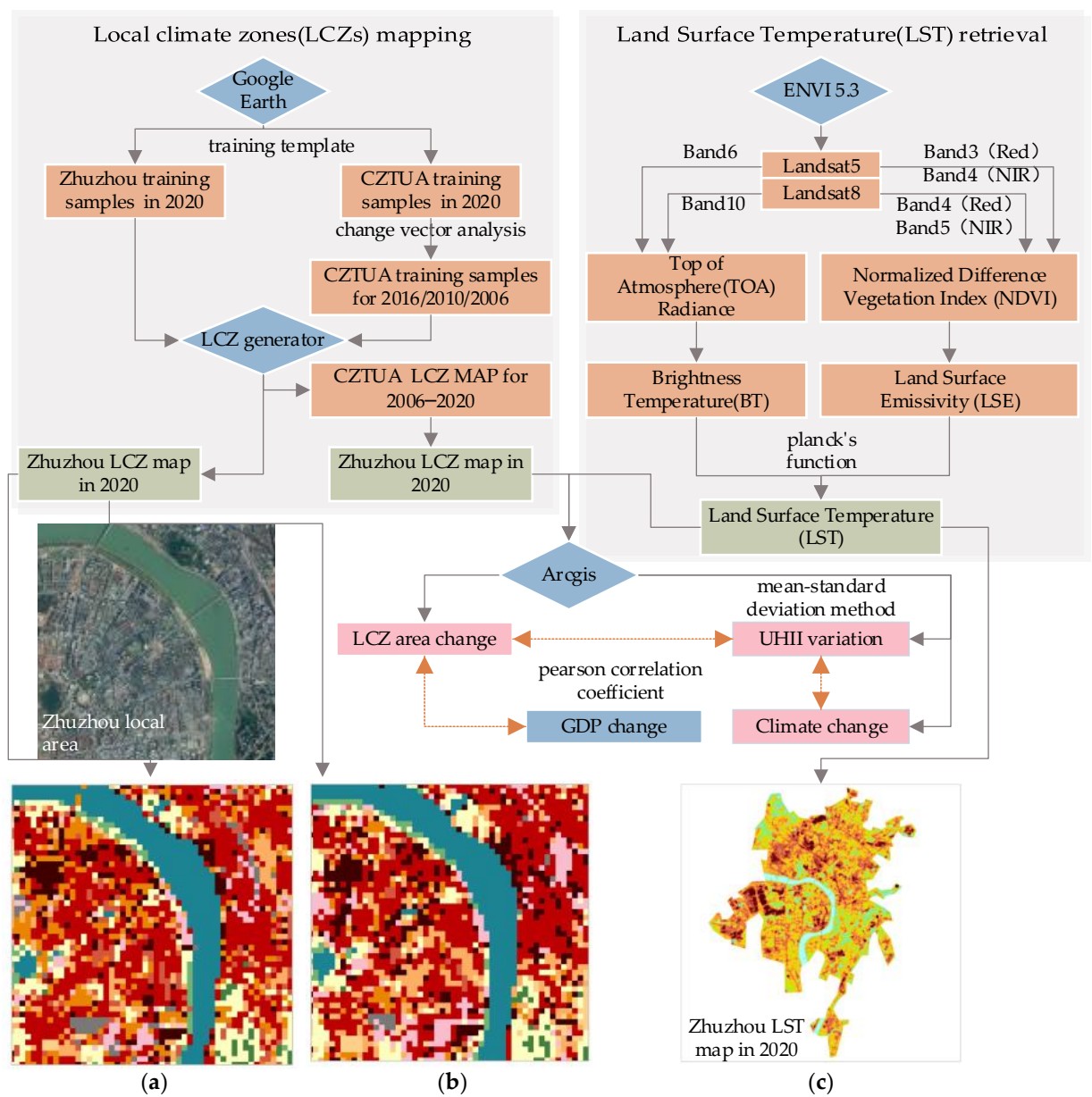

**Figure 2.** Flowchart for generating the LCZ map and LST map, taking Zhuzhou 2020 as an example. (**a**) The local LCZ map generated by the traditional method. (**b**) The local LCZ map generated by the improved method. (**c**) The LST map of Zhuzhou in 2020.

**Table 1.** Images from Google Earth and street view of LCZs.

| LCZ Classes | Google Earth | Street View | Representative Area |
|---|---|---|---|
| LCZ1 compact high rise | | | central business district, high-rise residential, apartment tower |
| LCZ2 compact mid rise | | | business district, old town |

**Table 1.** *Cont.*

| LCZ Classes | Google Earth | Street View | Representative Area |
|:---:|:---:|:---:|:---:|
| LCZ3 compact low rise | | | high-density terrace, old or dense town, village |
| LCZ4 openhigh rise | | | multistorey tenement |
| LCZ5 open mid rise | | | multiunit housing, research/business park, campus |
| LCZ6 open low rise | | | small retail shop, low-density terrace/row housing |
| LCZ8 large low rise | | | modern warehouse, storage facility |
| LCZ9 sparsely built | | | farm, country estate, low-density suburb |
| LCZ10 heavy industry | | | refinery, factory |
| LCZA dense trees | | | dense plantation, forest park |
| LCZD low plants | | | farmland, green space |
| LCZG water | | | urban river, lake |

2.  Improved WUDAPT method

The improved WUDAPT method is used in this paper to make training areas for the three cities in 2006, 2010, 2016, and 2020. The approach comprises the four major phases listed below:

1.  For the digitization of the training area for 2020 in Google Earth, we used the training template provided by WUDAPT in Google Earth [47]. Representative zones were selected for each LCZ type and used as training areas. We classified the CZT urban

agglomeration in 2020 to obtain training areas for the three cities simultaneously [19]. Each LCZ contains approximately 40 polygons.

2.  To generate training areas for 2016, 2010, and 2006 by using the time series sample transfer method, we first used a change vector analysis to assess whether a pixel has changed between 2020 and 2016, using the multispectral and thermal infrared bands of the Landsat images. We calculated the gray value of a pixel in 2020 and 2016 and then calculated the change vector and the change magnitude. When the change magnitude exceeds a certain threshold, the pixel is identified as a changed pixel. Second, the reliability of the pixels was checked through the specificity of a probability distribution. Finally, based on invariability and reliability, the k-nearest neighbor algorithm is used to further select representative training areas. To obtain the CZT urban agglomeration training areas for 2016, we deleted the training areas whose pixels changed from 2020 to 2016 and added new training areas. The training areas of 2010 and 2006 were obtained by using the same method [23,53].

3.  To obtain the LCZ map of the CZT urban agglomeration, we submitted selected training areas of the CZT urban agglomeration for 2006, 2010, 2016, and 2020 to the LCZ generator, respectively. After successful submission, the LCZ classification and quality control procedures were carried out in the generator to generate reliable LCZ maps. Metadata and unreliable polygons were marked. Next, the LCZ maps of the CZT urban agglomeration were obtained.

4.  In ArcGIS, the LCZ maps of the three cities for 2006, 2010, 2016, and 2020 were extracted from the LCZ maps of the CZT urban agglomeration by using the boundary of the study areas.

3.  Verification of the accuracy of the LCZ map

To ensure the quality of the generated LCZ maps, the WUDAPT applies an automatic cross-validation approach using 25 bootstraps. In each bootstrap, 70% of the training area polygons are used for training, and 30% are used for testing. The polygons are chosen by stratified (LCZ-type) random sampling while the original LCZ class frequency distribution is kept. This process is repeated 25 times to provide confidence intervals for the accuracy measures. In addition, this approach also allows for the creation of a probability map that represents how often (in percentage) a pattern was mapped in the iterative procedure. The resulting LCZ maps are based on all training areas, and the overall accuracy indicates the percentage of correctly classified pixels. The improved WUDAPT method increased the overall accuracy of the Zhuzhou LCZ classification in 2020 from 53% to 70%, according to a report by WUDAPT (Figure A1 in the Appendix A). Compared with generating an LCZ map of a city alone, its results are more reliable and accurate.

### 2.2.2. Generation of the LST Map

We obtain the remote-sensing images from the geospatial data cloud (http://www.gscloud.cn/, accessed on 9 September 2022). On the basis of cloud cover and availability, we select Landsat 5 images for 2006 and 2010 and Landsat 8 images for 2016 and 2020 (Table 2).

**Table 2.** Satellite imagery used in this study.

| Satellite | Landsat Entity ID | Acquisition Date | Cloud Cover |
| --- | --- | --- | --- |
| Landsat 5 | LT51230402006305BJC01 | 1 November 2006 | 0.00% |
| Landsat 5 | LT51230412006305BJC01 | 1 November 2006 | 0.00% |
| Landsat 5 | LT51230402010316BJC00 | 2 November 2010 | 0.00% |
| Landsat 5 | LT51230412010316BJC00 | 2 November 2010 | 1.00% |
| Landsat 8 | LC81230402016333LGN00 | 28 November 2016 | 3.10% |
| Landsat 8 | LC81230412016333LGN00 | 28 November 2016 | 0.61% |
| Landsat 8 | LC81230402020296LGN00 | 22 October 2020 | 0.02% |
| Landsat 8 | LC81230412020296LGN00 | 22 October 2020 | 0.03% |

1. The radiative transfer equation method

   In this paper, we use the radiative transfer equation method [54]. The process of retrieving the LST is as follows: First, Envi5.3 software is used to perform the radiometric correction, the atmospheric correction, the geometric correction, and the alignment on the image, and the study area is obtained by stitching and cropping. Second, the vegetation coverage, surface-specific emissivity, and blackbody radiance value are calculated at the same temperature. Finally, the LST is obtained by using the inverse function of Planck's formula.

2. Verification of the accuracy of the LST map

   The recorded temperatures of the three cities in 2006, 2010, 2016, and 2020 are obtained through the China Weather 5 platform (https://www.tianqi5.cn/, accessed on 18 September 2022). The reliability of the LST results is verified by comparing the average temperature between the recorded temperature data and the LST values retrieved via remote sensing. The average difference between the two sets of data is found to be less than 1 °C (Table 3).

**Table 3.** The recorded temperature data and the LST values.

| City Name | Date | Recorded Temperature (°C) | LST Value (°C) | Difference (°C) |
|---|---|---|---|---|
| Changsha | 1 November 2006 | 24.90 | 24.70 | 0.20 |
| Zhuzhou | 1 November 2006 | 24.60 | 25.30 | 0.70 |
| Xiangtan | 1 November 2006 | 25.20 | 25.09 | 0.11 |
| Changsha | 2 November 2010 | 21.10 | 22.83 | 1.73 |
| Zhuzhou | 2 November 2010 | 21.50 | 22.76 | 1.26 |
| Xiangtan | 2 November 2010 | 20.70 | 22.13 | 1.43 |
| Changsha | 28 November 2016 | 15.40 | 14.82 | 0.58 |
| Zhuzhou | 28 November 2016 | 15.90 | 15.41 | 0.49 |
| Xiangtan | 28 November 2016 | 15.60 | 15.12 | 0.48 |
| Changsha | 22 October 2020 | 24.00 | 25.44 | 1.44 |
| Zhuzhou | 22 October 2020 | 24.00 | 25.57 | 1.57 |
| Xiangtan | 22 October 2020 | 25.00 | 25.42 | 0.42 |

### 2.2.3. Extraction and Analysis

1. Extracting the area and LST of LCZs

   We extract the area of each LCZ in the CZT urban agglomeration and carry out a change-detection analysis that is based on LCZs' transition matrices by using the spatial analyst and data management tools in ArcGIS. The average LST values of LCZs are obtained by using the zonal statistics tools in ArcGIS.

2. Classifying LST by using the mean standard deviation method

   The advantage of using the mean and standard deviation is that it shows the spatial differences in LST regardless of the temporal variation in the actual LST value [19]. First, we calculate the average LST of the CZT urban agglomeration. Second, we use the standard deviation method to classify the LST values into seven categories: extremely high temperature, moderately high temperature, relatively high temperature, medium temperature, relatively low temperature, moderately low temperature, and extremely low temperature [22].

### 2.2.4. Linear Correlation Analysis

In this paper, we take the LCZ area data, the LST data, and the GDP data (Table 4) of the three cities as objects and use the analysis tools in SPSS to analyze the correlation of each factor. We build three models. The first variable of each model is the explanatory variable, and the second is the response variable: Model 1, the LCZ area and LST; Model 2, the weather temperature and the LST difference; and Model 3, the LCZ area change rate, and the GDP change rate. The weather temperature refers to the average LST values of

the CZT urban agglomeration in 2006, 2010, 2016, and 2020. The LST difference refers to the difference in the average LST between LCZs and a city. We use the LST difference to reveal the warming and cooling effects of LCZs. The calculation method for the LCZ area change rate is to divide the change in the LCZ area by the initial LCZ area. For example, the difference obtained by subtracting the LCZ 1 area in 2006 from that in 2010 is divided by the total area of LCZ 1 in 2006, and this value is the LCZ 1 area change rate from 2006 to 2010. The GDP includes the gross production values of the primary industry, secondary industry, and tertiary industry. The calculation method for the GDP is to divide the change in GDP by the initial GDP. The Pearson correlation analysis model is applied. The Pearson correlation coefficient r is widely applied in building science research to assess the strength of a two-variable linear association [23,55–57], and its value ranges from −1 to 1. When r is closer to 1, it reveals a stronger positive correlation between two variables, and when r is closer to −1, it indicates a stronger negative correlation. The *p*-value is used to test the significance of the correlation. When $p < 0.05$, there is a significant correlation, and when $p < 0.01$, there is a very significant correlation. Finally, we adjust the *p*-values by using the false-discovery-rate method [58].

**Table 4.** The GDP data for Changsha, Zhuzhou, and Xiangtan.

| City | Changsha | | | | Zhuzhou | | | | Xiangtan | | | |
|---|---|---|---|---|---|---|---|---|---|---|---|---|
| Year | 2006 | 2010 | 2016 | 2020 | 2006 | 2010 | 2016 | 2020 | 2006 | 2010 | 2016 | 2020 |
| Primary industry | 123 | 202 | 371 | 423 | 76 | 124 | 197 | 256 | 61 | 96 | 151 | 169 |
| Secondary industry | 791 | 2437 | 4513 | 4739 | 312 | 737 | 1318 | 1437 | 192 | 499 | 976 | 1175 |
| Tertiary industry | 885 | 1908 | 4473 | 6980 | 218 | 415 | 973 | 1413 | 169 | 299 | 740 | 999 |
| GDP (CNY 100 million) | 1799 | 4547 | 9357 | 12,143 | 605 | 1275 | 2488 | 3106 | 422 | 894 | 1867 | 2343 |

## 3. Results

### 3.1. LCZ Classification Results

3.1.1. Distribution and Proportion of LCZ

As is shown in Figure 3, we identified nine built-up land types (LCZs 1–6 and 8–10) and three natural land types (LCZs A, D, and G) in the CZT urban agglomeration. The compact buildings (LCZs 1–3) and open buildings (LCZs 4–6) are highly concentrated on both sides of LCZ G (water) in the three cities. LCZ 9 (sparsely built) and LCZ D (low plants) surround them. LCZ 8 (large low rise) is dispersed along the study areas' margins. In Changsha, LCZs 1–3 are centered in Furong Central Square and Yuelu Avenue, LCZ 8 is in the North Hub, LCZ A (dense trees) is in the Yuelu Mountain and Gushan Mountain areas, and LCZ G is mainly in the Xiangjiang River area. In Zhuzhou, LCZ 1–3 are concentrated in the Zhuzhou Railway Station and the Zhuzhou municipal government location, LCZ 8 is in the Tianyuan Industrial Area, LCZ A is in the Shifeng Mountain area, and LCZ G is mainly in the Xiangjiang River area. In Xiangtan, LCZs 1–3 are concentrated in Jinxiangtan Commercial Plaza, the municipal government location, and Baiyun Plaza; LCZ G is mainly in the Xiangjiang River area, and there is no obvious distribution of LCZ A. The distribution areas of LCZ 1 (compact high rise) and LCZ 8 in the CZT urban agglomeration have clearly expanded from 2006 to 2020. In the three cities, the study area is dominated by built-up types (Table 5). The largest proportion of built-up types is LCZ 9, followed by LCZ 8 and LCZ 2 (compact mid rise). In 2006 and 2010, the proportion of LCZ 1 is the smallest. In 2016 and 2020, the proportion of LCZ 10 (heavy industry) is the smallest. Changsha has the largest proportion of compact buildings, followed by Zhuzhou. Xiangtan has the largest proportion of LCZ D and LCZ G. Among the three cities, Changsha has the largest proportion of LCZ A, followed by Zhuzhou and Xiangtan.

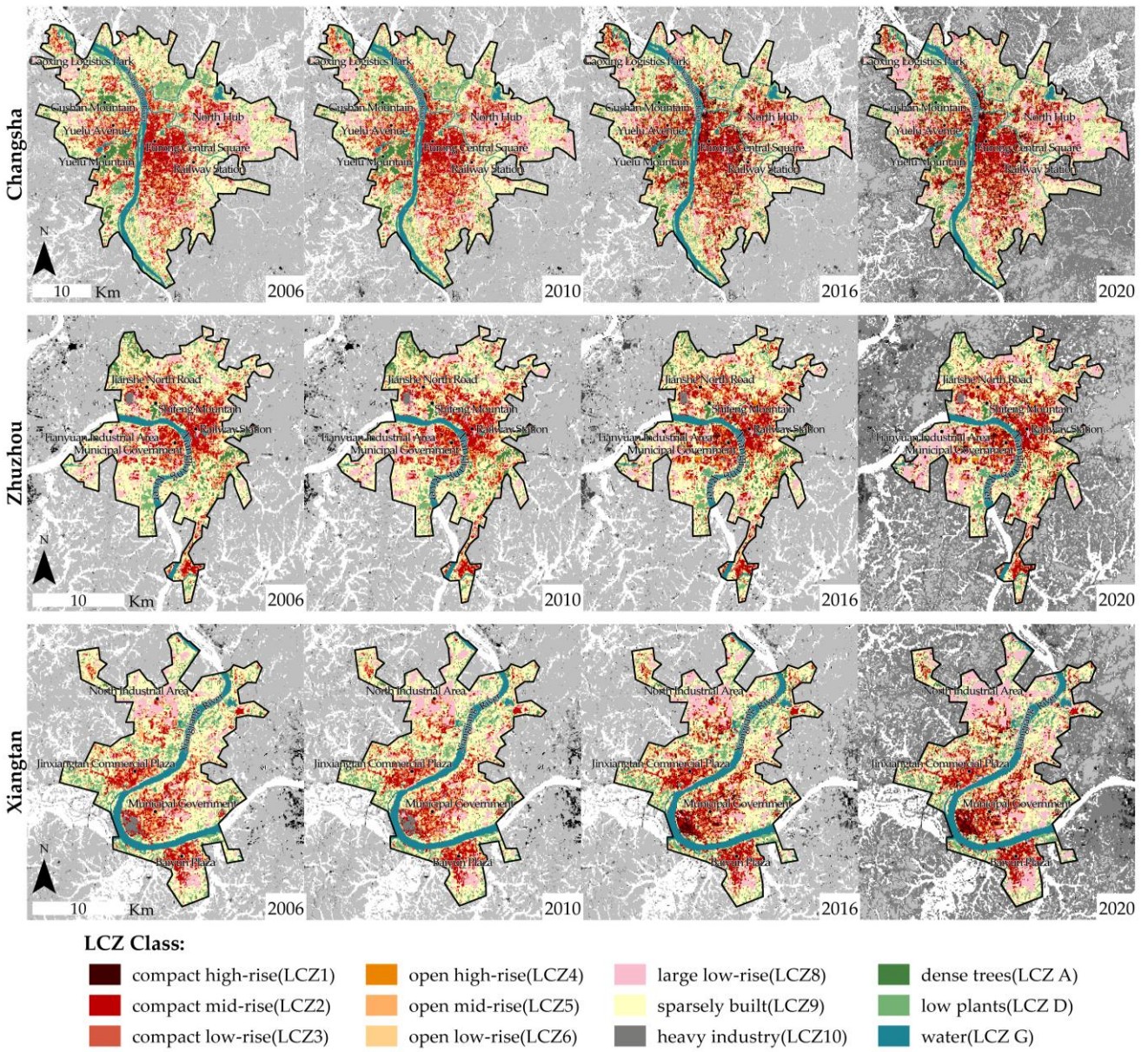

**Figure 3.** LCZ maps of Changsha, Zhuzhou, and Xiangtan in 2006, 2010, 2016, and 2020.

**Table 5.** The LCZ proportions of Changsha, Zhuzhou, and Xiangtan in 2006, 2010, 2016, and 2020.

| LCZ Proportion (%) | 2020 | | | 2016 | | | 2010 | | | 2006 | | |
|---|---|---|---|---|---|---|---|---|---|---|---|---|
| | Changsha | Zhuzhou | Xiangtan | Changsha | Zhuzhou | Xiangtan | Changsha | Zhuzhou | Xiangtan | Changsha | Zhuzhou | Xiangtan |
| LCZ 1 | 7.32 | 4.33 | 5.12 | 5.34 | 2.85 | 3.99 | 0.68 | 0.22 | 0.27 | 0.24 | 0.06 | 0.05 |
| LCZ 2 | 12.18 | 14.13 | 11.28 | 13.33 | 14.86 | 12.41 | 15.13 | 15.18 | 11.99 | 14.57 | 14.16 | 11.11 |
| LCZ 3 | 1.30 | 1.44 | 1.34 | 1.35 | 1.60 | 1.65 | 1.25 | 1.90 | 1.62 | 3.17 | 2.57 | 2.68 |
| LCZ 4 | 1.48 | 1.82 | 1.04 | 1.75 | 2.53 | 1.44 | 0.41 | 0.95 | 0.22 | 0.24 | 0.23 | 0.10 |
| LCZ 5 | 4.83 | 5.37 | 5.05 | 5.54 | 6.05 | 6.06 | 7.57 | 8.27 | 8.13 | 6.59 | 8.75 | 7.22 |
| LCZ 6 | 4.55 | 5.41 | 4.06 | 4.68 | 5.15 | 3.97 | 4.96 | 2.69 | 2.96 | 3.21 | 3.02 | 2.64 |
| LCZ 8 | 19.29 | 17.76 | 19.99 | 14.32 | 11.35 | 12.82 | 13.56 | 13.45 | 12.84 | 11.64 | 11.35 | 11.19 |
| LCZ 9 | 30.04 | 33.64 | 31.48 | 34.15 | 39.50 | 36.52 | 36.31 | 41.09 | 40.53 | 39.88 | 42.92 | 43.19 |
| LCZ 10 | 0.10 | 0.48 | 0.19 | 0.35 | 0.65 | 0.41 | 0.72 | 0.83 | 1.29 | 0.40 | 0.81 | 1.10 |
| LCZ A | 4.90 | 4.42 | 1.02 | 4.92 | 4.34 | 1.04 | 4.83 | 4.49 | 1.02 | 4.93 | 4.72 | 1.15 |
| LCZ D | 7.46 | 6.16 | 9.77 | 7.71 | 6.10 | 10.06 | 8.39 | 6.15 | 9.87 | 8.94 | 6.66 | 10.36 |
| LCZ G | 6.54 | 5.03 | 9.64 | 6.54 | 4.94 | 9.60 | 6.19 | 4.74 | 9.24 | 6.19 | 4.74 | 9.19 |

### 3.1.2. Area Changes in LCZs

As is shown in Figure 4a, from 2006 to 2020, the total area of LCZ 8 (large low rise) increased the most, followed by LCZ 1 (compact high rise) in the CZT urban agglomeration.

In Changsha, LCZ 9 (sparsely built) showed the largest area reduction, followed by LCZ 2 (compact mid rise). In Zhuzhou and Xiangtan, LCZ 9 had the largest reduction in area, followed by LCZ 5 (open mid rise). The area of LCZ A (dense trees) and LCZ G (water) in the CZT urban agglomeration largely remained unchanged. As is shown in Figure 4b, from 2006 to 2010, LCZ 1 had the largest growth rates among the three cities. Its growth almost stagnated from 2016 to 2020. The growth rate of LCZ 4 (open high rise) is second, and its changing pattern is consistent with LCZ 1. From 2006 to 2010, LCZ 2 in Changsha rapidly decreased. From 2010 to 2020, LCZ 10 (heavy industry) in the CZT urban agglomeration also rapidly decreased.

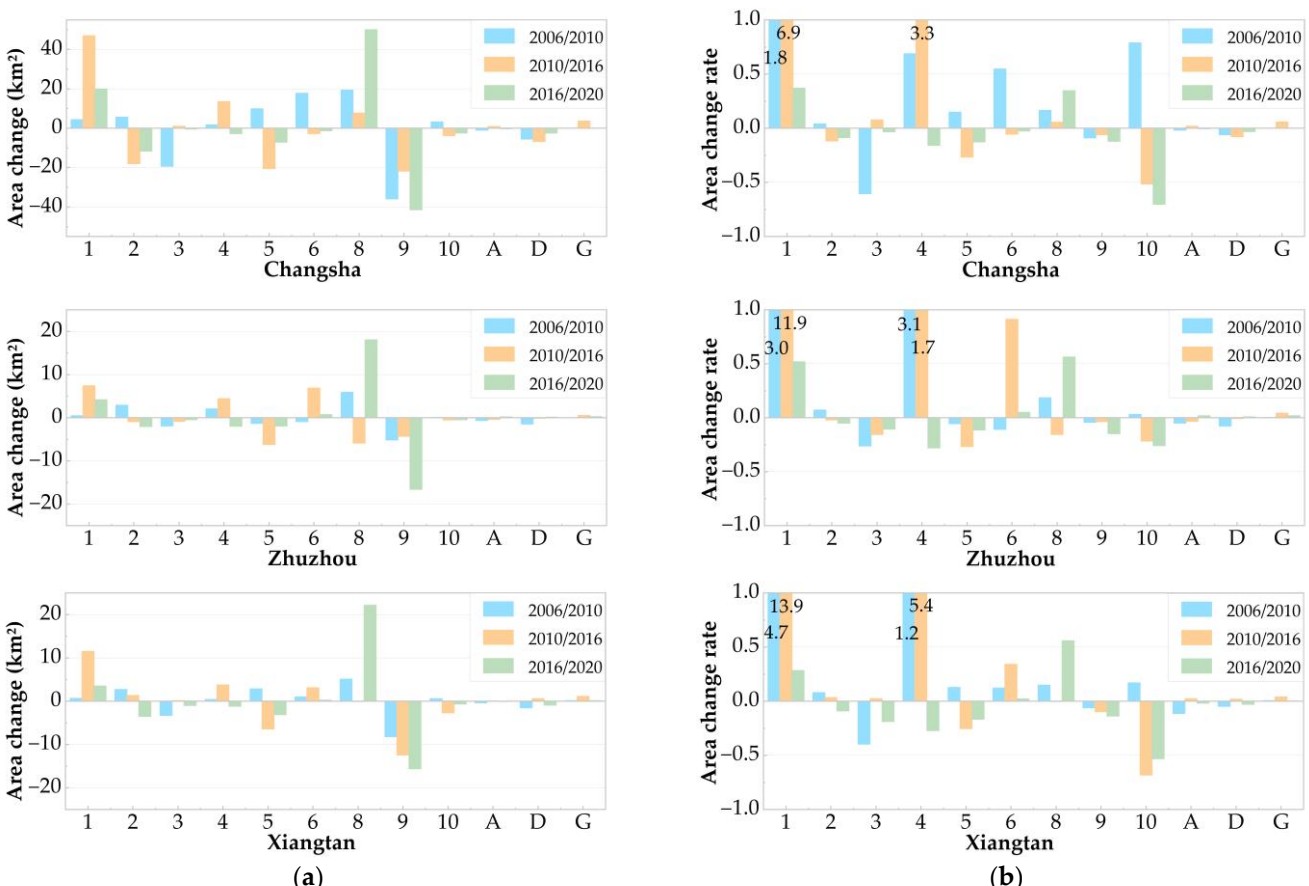

**Figure 4.** Area changes in the LCZs of Changsha, Zhuzhou, and Xiangtan for 2006–2010, 2010–2016, and 2016–2020. (**a**) The LCZ area change (km²). (**b**) The LCZ area change rate.

From 2006 to 2020, in the CZT urban agglomeration, the transfer between LCZ 2/LCZ 1, LCZ 5/LCZ 2, LCZ 9/LCZ 6 (open low rise), LCZ 9/LCZ 8, and LCZ 9/LCZ D (low plants) are close. Between them, LCZ 9 has the largest area converted into LCZ 8 (Figure 5).

### 3.2. LST Distribution Results

As is shown in Figure 6, the extremely high and moderately high temperature areas of the CZT urban agglomeration were distributed mainly along the Xiangjiang River in 2006, which spread to the surrounding areas by 2010 and which spread to the entire study areas by 2016. There are several obvious concentration regions for extremely and moderately high temperatures. In Changsha, the extremely and moderately high temperatures are distributed mostly in the North Hub and Changsha Railway Station in 2006 and 2010, and in 2016 and 2020, they are distributed mainly in the North Hub, Yuelu Avenue, and Gaoxing Logistics Park. The moderately low and extremely low temperatures are dispersed primarily in the Yuelu Mountains, the Gushan Mountains, and Xiangjiang River area from 2006 to 2020. In Zhuzhou, the moderately and extremely high temperatures are concen-

trated along Jianshe North Road, at the Zhuzhou Railway Station, and in the Zhuzhou municipal government location in 2006 and 2010, with the addition of the concentration region of the Tianyuan Industrial Area in 2016 and 2020. From 2006 to 2020, the moderately and extremely low temperatures were concentrated in the Xiangjiang River and Shifeng Mountain. From 2006 to 2020, the moderately and extremely high temperatures in Xiangtan are concentrated in the North Industrial Area, Jinxiangtan Commercial Plaza, the Xiangtan municipal government location, and Baiyun Plaza. The moderately and extremely low temperatures are concentrated in the Xiangjiang River.

There are obvious differences in the LST for each LCZ (Table 6). From 2006 to 2020, LCZ 10 (heavy industry) had the highest LST, followed by LCZ 8 (large low rise) in the CZT urban agglomeration. LCZ 2 (compact mid rise) and LCZ 3 (compact low rise) had higher LSTs than LCZ 5 (open mid rise) and LCZ 6 (open low rise). LCZ G (water) and LCZ A (dense trees) had the lowest LST values. The extremely high to moderately high temperatures were distributed in LCZ 8 and LCZ 10 (Figure 7). The moderately low to extremely low temperatures were distributed in LCZ A and LCZ G. The cooling effect of LCZ G was very significant, followed in significance by that of LCZ A. The LST difference of high-rise buildings (LCZ 1, 4) is lower than that of low-rise buildings (LCZ2–3, 5–6). From 2010 to 2020, the LST differences of high-rise buildings were negative in Changsha and in Zhuzhou.

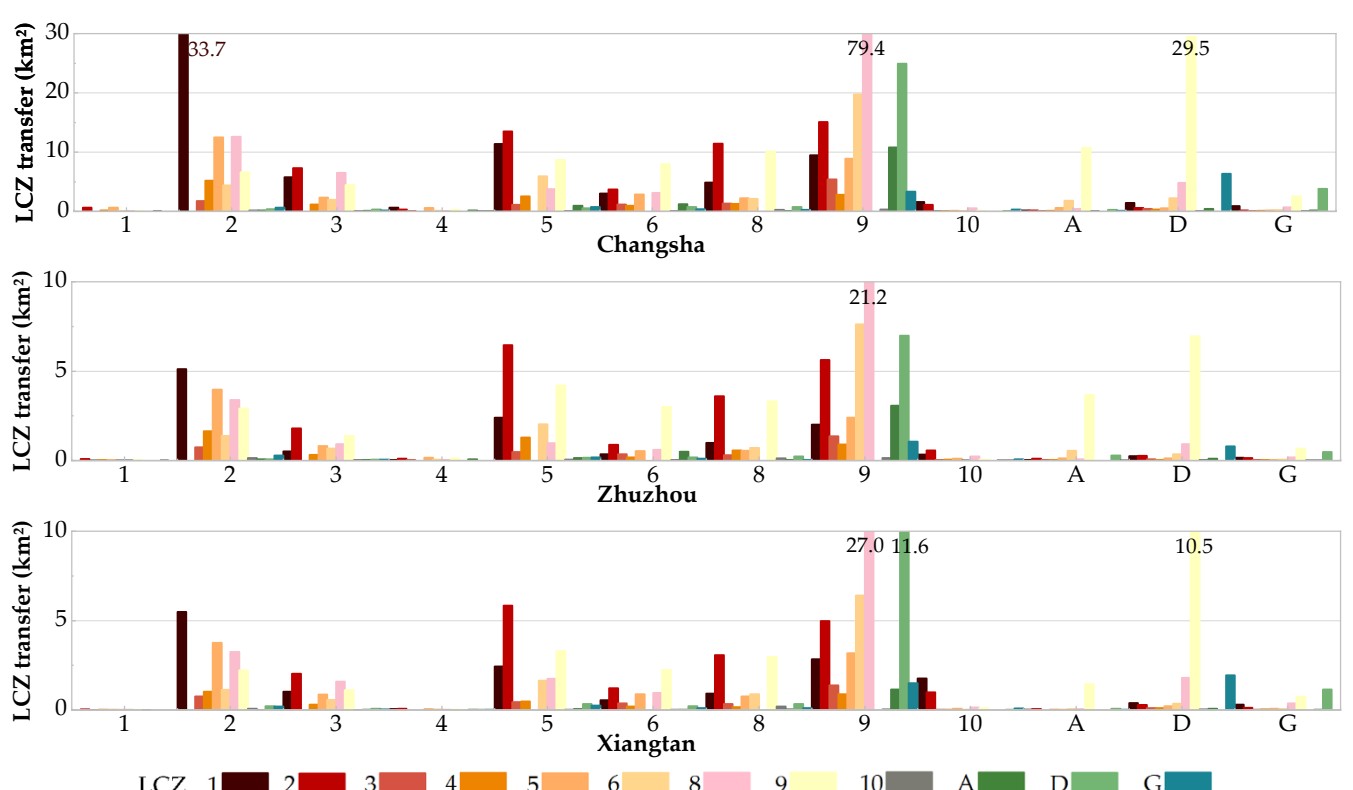

**Figure 5.** LCZ transfers, 2006–2020 (km$^2$). The horizontal axis represents LCZs, and the vertical axis represents the transfer area from 2006 to 2020.

**Table 6.** Average LST of each LCZ.

| LCZ Class (°C) | 2006 | | | 2010 | | | 2016 | | | 2020 | | |
|---|---|---|---|---|---|---|---|---|---|---|---|---|
| | Changsha | Zhuzhou | Xiangtan | Changsha | Zhuzhou | Xiangtan | Changsha | Zhuzhou | Xiangtan | Changsha | Zhuzhou | Xiangtan |
| LCZ 1 | 25.34 | 25.98 | 26.18 | 22.30 | 22.28 | 22.79 | 14.60 | 15.59 | 15.97 | 25.12 | 25.34 | 26.12 |
| LCZ 2 | 25.77 | 26.55 | 26.48 | 23.70 | 23.90 | 23.09 | 15.68 | 16.34 | 15.92 | 26.36 | 26.41 | 26.51 |
| LCZ 3 | 25.73 | 26.36 | 26.28 | 23.67 | 23.86 | 23.24 | 15.78 | 16.36 | 16.08 | 26.29 | 26.66 | 26.34 |
| LCZ 4 | 24.57 | 24.96 | 25.62 | 21.98 | 22.10 | 21.84 | 14.85 | 15.55 | 15.38 | 25.04 | 25.20 | 25.27 |

**Table 6.** *Cont.*

| LCZ Class (°C) | 2006 | | | 2010 | | | 2016 | | | 2020 | | |
|---|---|---|---|---|---|---|---|---|---|---|---|---|
| | Changsha | Zhuzhou | Xiangtan | Changsha | Zhuzhou | Xiangtan | Changsha | Zhuzhou | Xiangtan | Changsha | Zhuzhou | Xiangtan |
| LCZ 5 | 25.20 | 26.10 | 26.09 | 23.17 | 23.55 | 22.68 | 15.23 | 15.96 | 15.47 | 25.60 | 25.80 | 25.75 |
| LCZ 6 | 24.86 | 25.40 | 25.73 | 22.97 | 22.31 | 22.39 | 15.22 | 15.73 | 15.62 | 25.47 | 25.46 | 25.63 |
| LCZ 8 | 25.45 | 25.73 | 25.60 | 24.09 | 23.93 | 23.33 | 16.72 | 17.21 | 16.88 | 27.22 | 27.29 | 27.07 |
| LCZ 9 | 24.51 | 24.99 | 24.99 | 22.72 | 22.51 | 22.15 | 15.22 | 15.73 | 15.50 | 25.19 | 25.13 | 25.07 |
| LCZ 10 | 25.92 | 28.26 | 28.11 | 23.37 | 25.23 | 24.26 | 15.83 | 18.71 | 16.83 | 27.13 | 28.54 | 28.28 |
| LCZ A | 22.60 | 23.84 | 23.80 | 20.36 | 20.79 | 20.48 | 13.84 | 14.53 | 14.39 | 23.38 | 23.42 | 23.61 |
| LCZ D | 24.54 | 24.84 | 24.77 | 22.72 | 22.57 | 22.03 | 15.27 | 15.86 | 15.53 | 24.94 | 24.97 | 24.85 |
| LCZ G | 22.88 | 22.32 | 22.29 | 20.03 | 18.88 | 18.55 | 13.87 | 14.36 | 14.16 | 21.62 | 21.88 | 21.65 |

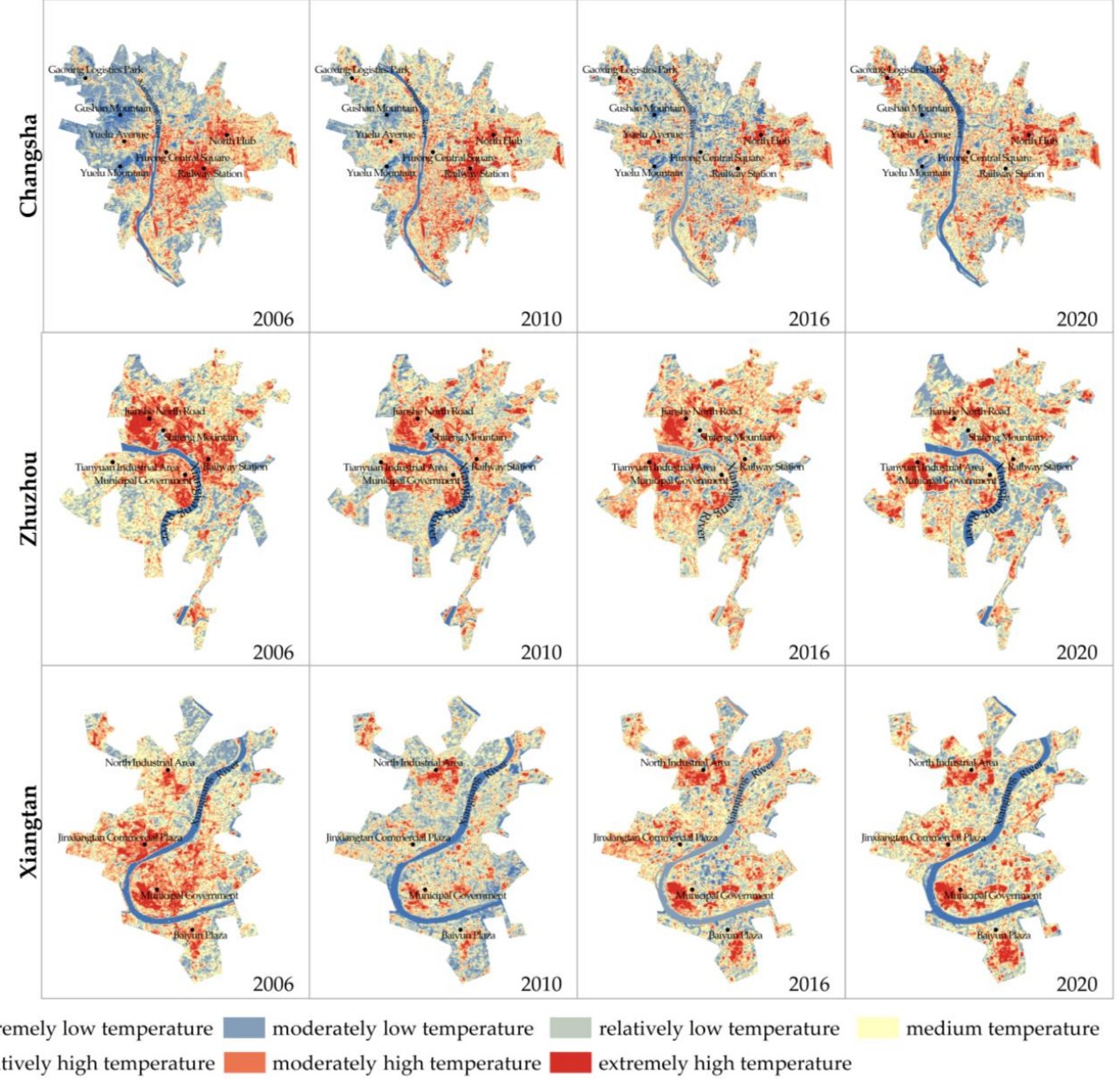

**Figure 6.** The LST spatial distribution of Changsha, Zhuzhou, and Xiangtan in 2006, 2010, 2016, and 2020, respectively.

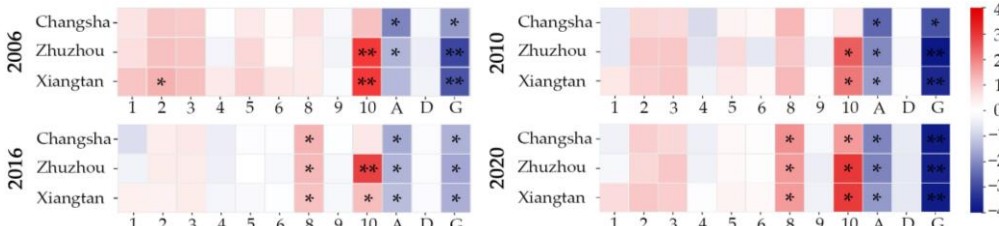

**Figure 7.** Land surface temperature (LST) differences between LCZs and the cities of Changsha, Zhuzhou, and Xiangtan in 2006, 2010, 2016, and 2020. Notes: * a moderately high or moderately low temperature; ** an extremely high or extremely low temperature.

*3.3. Correlation Analysis Results*

The relationship between the LCZ area and its LST shows no significant correlation between the total areas of LCZ and LST (Table 7). The LST differences in LCZ 2, LCZ 3, LCZ D, and LCZ G have very significant positive correlations with the weather temperature. The LST difference in LCZ 5 has a significant positive correlation with the weather temperature. The LST differences in LCZ 1, LCZ 4, LCZ 6, LCZ 8–10, and LCZ A are not significantly correlated with the weather temperature. The growth rate of the tertiary industry GDP has a very significant positive correlation with the area change rate of LCZ 1 (compact high rise), a significant positive correlation with the area change rate of LCZ 4 (open high rise), and a very significant negative correlation with the area change rate of LCZ 8 (large low rise). The growth rate of the secondary industry GDP has a very significant positive correlation with LCZ 10 (heavy industry), a significant positive correlation with LCZ 2 (compact mid rise), and a significant negative correlation with LCZ 3 (compact low rise). The growth rate of the regional primary industry GDP has a significant positive correlation with LCZ 9 (sparsely built) and a significant negative correlation with LCZ 8.

**Table 7.** Pearson correlation coefficient among variables.

| Variable | LCZ 1 | LCZ 2 | LCZ 3 | LCZ 4 | LCZ 5 | LCZ 6 | LCZ 8 | LCZ 9 | LCZ 10 | LCZ A | LCZ D | LCZ G |
|---|---|---|---|---|---|---|---|---|---|---|---|---|
| Area | −0.474 | −0.161 | 0.051 | −0.304 | −0.125 | −0.044 | 0.398 | 0.298 | −0.556 | −0.537 | 0.266 | 0.249 |
| | (0.120) | (0.617) | (0.875) | (0.337) | (0.699) | (0.893) | (0.211) | (0.346) | (0.061) | (0.072) | (0.403) | (0.435) |
| | (0.300) | (0.506) | (0.593) | (0.402) | (0.538) | (0.598) | (0.361) | (0.404) | (0.270) | (0.270) | (0.415) | (0.420) |
| Weather temperature | 0.415 | 0.901 ** | 0.902 ** | 0.230 | 0.691 * | 0.386 | −0.131 | −0.492 | 0.434 | 0.516 | −0.775 ** | −0.758 ** |
| | (0.180) | (0.000) | (0.000) | (0.473) | (0.013) | (0.215) | (0.684) | (0.104) | (0.159) | (0.086) | (0.003) | (0.004) |
| | (0.063) | (0.000) | (0.000) | (0.145) | (0.008) | (0.071) | (0.197) | (0.046) | (0.059) | (0.040) | (0.003) | (0.003) |
| Primary industry | 0.547 | 0.387 | −0.118 | 0.678 | 0.043 | 0.251 | −0.768 * | 0.799 * | 0.394 | −0.133 | −0.443 | 0.531 |
| | (0.127) | (0.304) | (0.762) | (0.045) | (0.912) | (0.515) | (0.016) | (0.010) | (0.294) | (0.733) | (0.232) | (0.141) |
| | (0.156) | (0.220) | (0.414) | (0.087) | (0.458) | (0.323) | (0.046) | (0.046) | (0.218) | (0.405) | (0.201) | (0.163) |
| Secondary industry | 0.193 | 0.757 * | −0.722 * | 0.346 | 0.638 | 0.309 | −0.523 | 0.613 | 0.818* | −0.501 | −0.485 | −0.134 |
| | (0.619) | (0.018) | (0.028) | (0.362) | (0.064) | (0.418) | (0.149) | (0.079) | (0.007) | (0.169) | (0.185) | (0.732) |
| | (0.218) | (0.033) | (0.035) | (0.144) | (0.054) | (0.159) | (0.080) | (0.059) | (0.026) | (0.084) | (0.086) | (0.248) |
| Tertiary industry | 0.842 * | 0.249 | 0.116 | 0.792 * | −0.285 | 0.547 | −0.916 ** | 0.657 | 0.098 | 0.170 | −0.047 | 0.685 |
| | (0.004) | (0.518) | (0.767) | (0.011) | (0.457) | (0.127) | (0.001) | (0.055) | (0.801) | (0.662) | (0.905) | (0.042) |
| | (0.014) | (0.434) | (0.532) | (0.026) | (0.404) | (0.158) | (0.007) | (0.079) | (0.543) | (0.495) | (0.573) | (0.068) |

Note: $p$-values are in the first parentheses, and the adjusted $p$-values are in the second parentheses; * and ** denote statistical significance at the 0.05 and 0.01 levels, respectively.

## 4. Discussion

The LCZ classification results of the CZT urban agglomeration show that Changsha has the largest proportion of high-density built-up types (LCZ 1–3). This is because Changsha is the most economically developed city in Hunan Province, and its urban construction level is higher than that of the other two cities. Xiangtan is the least economically developed city among the three cities [50]. Among the three cities, Changsha has the largest proportion of dense trees, followed by Zhuzhou and Xiangtan, in that order. This is because there

are two large mountains, Yuelu Mountain and Gushan Mountain, in the built-up area of Changsha. The mountains are densely vegetated. Shifeng Mountain is in Zhuzhou, but there is no large mountain in the built-up area of Xiangtan. The area changes in LCZ show that building construction in the CZT urban agglomeration tends to be high rise, dense, and industrial. The CZT urban agglomeration attaches great importance to protecting nature in that it has constructed "two-type" cities. Since 2006, Changsha has vigorously developed high-tech, new, and modern service industries according to the master plan. Zhuzhou has carried out the technological improvement and transformation of the environment for traditional industrial areas. Industries in the city's central area that cannot exert the benefits of land differentials and create pollution will be gradually closed and relocated. Xiangtan plans to become a strong industrial city dominated by "two-type" industries [59].

The LST difference in the LCZ indicates that it is appropriate to use the LCZ system to study the heat island effect. The LST is closely related to the characteristics of the type of surface cover. The large low-rise and heavy industries significantly increase the area temperature. The cooling effect of the water is very significant, followed by that of dense trees. This result is consistent with the results of Zhu and Dimitrov [7,26]. As building density increases, the warming effect intensifies. Mushore also came to the same conclusion [14]. However, the continuous increase in building height will not lead to a continuous increase in the regional warming effect. This may be because when the building reaches a certain height, the shadow of the building increases, and the solar radiation in the area decreases. High-rise buildings have greater airflow. This result contradicts Mushore [60]. The relationship between the LCZ area and LST shows that the change in building type does not increase the average LST in this area. Changes in the natural type (LCZ A, D, G) did not produce significant changes in the cooling effect. This is consistent with N. Li's conclusion [27]. The relationship between the LST difference and the weather temperature shows that the LST difference of the compact mid-rise, the compact low-rise, and the open mid-rise types significantly vary with the weather temperature, and the LST differences do not significantly change with a weather temperature change in high-rise, open low-rise, large low-rise, sparsely built, and heavy-industry types of areas. This may be due to the low air circulation and weak heat dissipation capacity of the compact mid-rise, compact low-rise, and open mid-rise types. The cooling capacity of high-rise and open low-rise buildings is stronger than that of compact mid-rise, low-rise, and open low-rise buildings. Large low-rise, sparsely built, and heavy-industry buildings are low in height and open, and they do not have a strong thermal insulation capacity. When the weather temperature changes, the LST differences in the low plants and water zones significantly change [61]. Therefore, under hot weather conditions, high-rise, open low-rise, sparsely built, low-plant, water, and dense-tree areas are more comfortable [62,63]. Compact mid-rise, compact low-rise, and open mid-rise built-up types very significantly respond to warming weather, and these areas are more uncomfortable when the weather is hot.

The results of the evolution of the LCZ and the development of urban GDP show that the construction of compact high rises has a very significant effect on the tertiary industrial economy. The construction of open high rises has a pulling effect on the tertiary industry's GDP. A significant portion of GDP in the CZT urban agglomeration is driven by tertiary industry [50]. It can be shown that compact high-rise and open high-rise buildings have a significant pulling effect on GDP. An increase in the large low-rise area has a very significant inhibitory effect on the development of tertiary industry. The GDP of the secondary industry comes from heavy industry. This is because the secondary industry is construction. Compact low rises significantly inhibit the development of secondary industry. This may be due to the low efficiency of construction land in compact low-rise areas, which is not conducive to increasing the value of the land. The development of the secondary industry requires much land. The growth rate of the primary industry has also rapidly slowed owing to the rapid reduction in sparsely built areas and the rapid increase in large low-rise areas. The primary industries are agriculture, forestry, animal husbandry,

and fishery. Sparsely built areas contain many low plants. Most of this land is used to construct large low-rise areas.

## 5. Conclusions

This study was based on Landsat satellite images and GDP data concerning the CZT urban agglomeration in 2006, 2010, 2016, and 2020. Using the WUDAPT and remote-sensing-inversion methods, the land cover and surface temperature of Changsha, Zhuzhou, and Xiangtan were classified, extracted, and analyzed. The following results were obtained:

1.  The LCZ map generated by the improved WUDAPT method has higher accuracy and efficiency than that created by using the traditional method, and the accuracy increased from 53% to 70%. From 2006 to 2020, the main built-up types in the CZT urban agglomeration were the sparsely built, large low-rise, and compact mid-rise types. Low-plant zones represent the largest proportion of the natural types, followed by water and dense-tree zones. Construction in the CZT urban agglomeration tends to involve high-rise, dense, and industrial buildings. Urban construction land is taken mainly from the sparsely built type of land.
2.  The LST values of the large low-rise and heavy-industry areas are significantly higher than the average LST values of the three cities. The LST values of the water and dense-tree areas are significantly lower than the average LST. Each LCZ has a stable LST, which has little correlation with the LCZ size.
3.  The LST differences between the compact mid-rise, compact low-rise, open mid-rise, low-plant, and water zones are significantly affected by changes in the weather temperature. The hotter the weather, the stronger the warming effect of the compact low rise. The warming effect of high-rise buildings does not significantly increase with a rise in temperature. When the weather becomes hotter, the cooling effect of dense-tree and water areas is more pronounced. Therefore, compact low rises are ineffective against climate warming and inhibit economic growth. Compact high rises and open high rises can not only effectively deal with climate warming but also significantly stimulate economic growth.

The results of this paper can help us to understand the heat island effect of land cover and its economic effect in the CZT urban agglomeration for the optimization of land-resource utilization. This is especially valuable in the absence of detailed studies on this topic. Being such a study is the main advantage of this paper. However, the shortcoming of this paper is that the weather temperature sample data are relatively limited. Adding weather temperature variables can improve the correlation analysis of LST with weather temperature changes in different LCZ types.

**Author Contributions:** Conceptualization and methodology, F.H. and L.L.; software, Y.H.; writing—original draft preparation, F.H. and L.L.; writing—review and editing, M.Z.; supervision, K.B.B. All authors have read and agreed to the published version of the manuscript.

**Funding:** This research was funded by the Education Bureau of Hunan Province, grant number 22B0285; the Key Disciplines of State Forestry Administration of China, grant number No. 21 of Forest Ren Fa, 2016; the Hunan Province "Double First-class" cultivation discipline of China, grant number No. 469 of Xiang Jiao Tong, 2018.

**Institutional Review Board Statement:** Not applicable.

**Informed Consent Statement:** Not applicable.

**Data Availability Statement:** The data are contained within the article. The data presented in this study are available on request from the corresponding author.

**Acknowledgments:** We thank all WUDAPT contributors for providing the training areas for our city of interest. Map data are copyrighted by OpenStreetMap contributors and are available from https://lcz-generator.rub.de/submissions (accessed on 15 September 2022).

**Conflicts of Interest:** The authors declare no conflict of interest.

## Appendix A

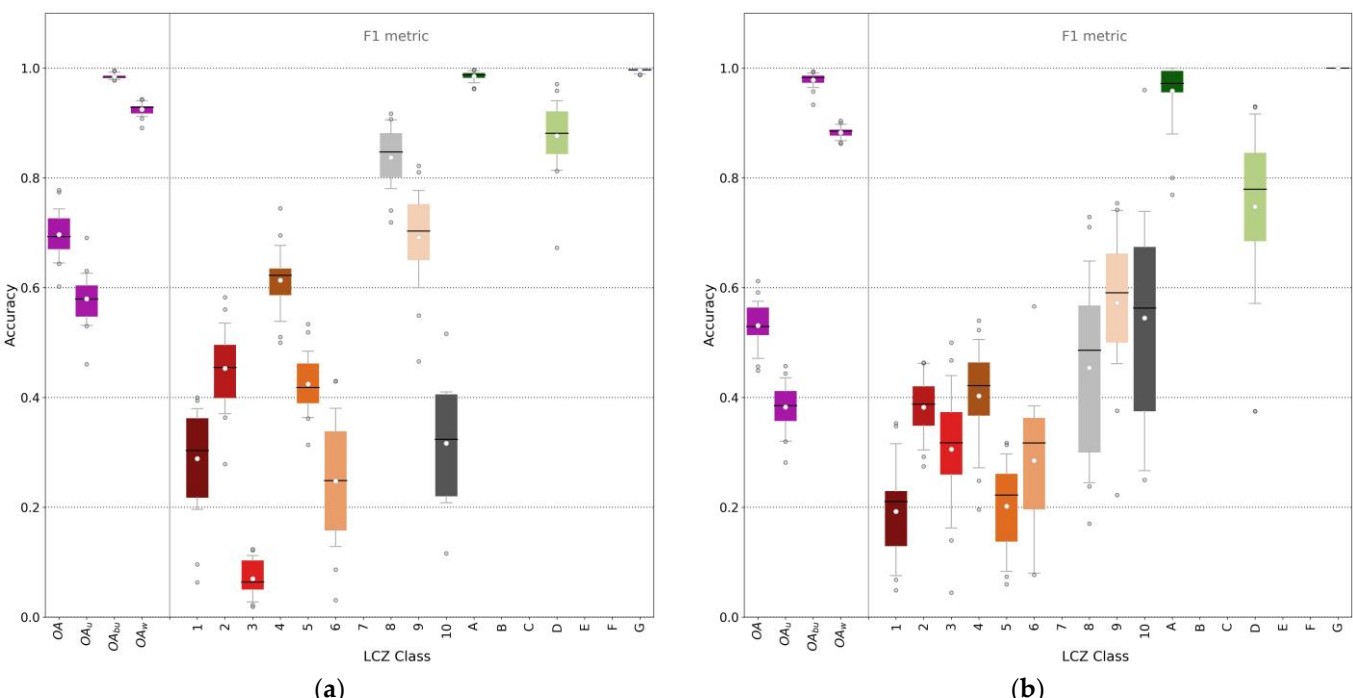

**Figure A1.** Accuracy reports sent by WUDAPT. (**a**) Improved method. (**b**) General method.

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
