# Peer review of "Investigating the Spatial Heterogeneity of Urban Heat Island Responses to Climate Change Based on Local Climate Zones"

_sustainability, doi:10.3390/su15076298_

Round 1

Reviewer 1 Report

The manuscript is supported by a sufficient amount of literature and a fairly expert, exact analysis of the problem, but it is reader-unfriendly. The first factor is an overabundance of abbreviations. I understand they can't be avoided, but why aren't they used uniformly? For most of the manuscript, the LCZ is referred to by its full name (e.g. LCZ G), but for example, in the conclusion, various LCZs are referred to, and LCZ G is referred to only as "G", which can be confusing for the reader. The second factor is the incompleteness of some sentences or not completely correct English, so these sentences are difficult to understand. The third and, in my opinion, the most significant problem is the fragmented division of the manuscript. Very often sentences that fit the methodology are given in the results, sentences that should be given in the results are found in the discussion, etc. To be more specific in my comments:

L64–67: The last sentences of the paragraph describe the aims of the manuscript, before the entire introduction is presented. They should be linked to the last paragraph: L95-L99, where the aims are formulated.

L229-L230: I don't understand if LCZ 1 or LCZ 10 had the lowest density in 2010 (which is repeated there twice).

L242-L243: What is the exact meaning of the sentence: LCZ 8 was the LCZ with the biggest increase (focusing on Changsha) or LCZ 8 grew the most in Changsha (compared to other cities) - the overall context of the results leads me to a different conclusion than the sentence implies grammatically. This is not the only strange sentence, but rather one of many examples and someone to smooth the text into a more reader-friendly version would be much needed.

L242: „It shows a significant increase“ – why is the meaning of the previous sentence so strangely repeated - was it not enough to connect the information about significance with the preceding sentence?

L254-L256: These sentences don't make much sense without the following sentences. However, I think that it would be better to connect the text organically so that it makes sense to the reader directly.

L276-L278. Study variables – are they explanatory variables or response variables? Which of the named is which. Why is UHII named twice? Is it because it appears as a variable in two different models (One with LCZ and the other with climate)? Why are these models not specified? Why is it stated in the results that there are some variables involved? This should be in the methodology. Why isn't it just written here if there was a significant effect between LCZ and UHII? And what does the sentence (without verbs) "The LCZ area change rate and city GDP growth rate" even mean?

L278-L279: „the standard regression coefficients, which reflect the correlation of two variables“: During the analysis, the authors have to decide whether there is a correlation between the two variables or a regression. I see no point in mixing these two analyses to analyze one set of variables (and write about them in the same sentence).

L294-L306: There is no such thing as very significant - the results are either significant or not significant, that's the point of a p-value. If the authors want to present proof to the readers that the p-value was really small, or relatively close to the 0.05 limit, then the entire manuscript lacks explicitly written p-values (and test values, degrees of freedom,...)

L354: Table 5. Why is this table (along with some graphs) even part of the discussion? Given that individual LCZs are correlated to some extent, it would be appropriate to adjust all p.values obtained by tests using adjustment (e.g. false discovery rate).

L361: „High-rise buildings (LCZ 1, 4) and open low-rise buildings (LCZ 6) are more capable of dissipating heat than they are“. This sentence doesn't make sense unless the authors meant something else by “they” - for example, the factors listed a few sentences back. In that case, instead of “they”, it was necessary to explicitly write "above-mentioned factors" or something similar.

L405: Why do LCZs (if several are listed consecutively) cease to be listed by their full name from this line and, inconsistently with the rest of the manuscript, are listed only by the final letter/number?

L411: ineffective rather than ineffectively? 

Author Response

Dear Reviewer,

We are grateful for your time and effort on reviewing the manuscript. The new manuscript has been revised based on your comments and the corrections made are explained below.

Hopefully, this revised version would be received favorably.

We are looking forward to hearing from you.

Sincerely yours,

Authors

☆ ☆ ☆ ☆ ☆

-Reviewer 1-

The manuscript is supported by a sufficient amount of literature and a fairly expert, exact analysis of the problem, but it is reader-unfriendly. The first factor is an overabundance of abbreviations. I understand they can't be avoided, but why aren't they used uniformly? For most of the manuscript, the LCZ is referred to by its full name (e. g. LCZ G), but for example, in the conclusion, various LCZs are referred to, and LCZ G is referred to only as "G", which can be confusing for the reader. The second factor is the incompleteness of some sentences or not completely correct English, so these sentences are difficult to understand. The third and, in my opinion, the most significant problem is the fragmented division of the manuscript. Very often sentences that fit the methodology are given in the results, sentences that should be given in the results are found in the discussion, etc. To be more specific in my comments:

L64–67: The last sentences of the paragraph describe the aims of the manuscript, before the entire introduction is presented. They should be linked to the last paragraph: L95-L99, where the aims are formulated.

Response:

Thank you very much for your suggestion, which helped us improve the coherence of this paper. We have adjusted L64-L67 to P3 L114-L121, which is related to the goals formulated in this paper, and we modified some expressions to connect the context. You can see it in detail in P3 L114-L121:

In view of this, we propose an improved WUDAPT method and take the LCZ system as the framework to analyze: 1. the correlation between LST and the total areas of LCZ; 2. under a high weather temperature, which LCZ types will be affected the most; and 3. which industries’ GDP will be impacted the most by LCZ changes. The purpose of this is to judge: 1. Whether the change in the total area of an LCZ type will affect the aver-age LST; 2. How the LST in different LCZ types will vary according to weather temperature changes; and 3. The economic benefits of each LCZ. Finally, we provide relevant recommendations for urban construction.

L229-L230: I don't understand if LCZ 1 or LCZ 10 had the lowest density in 2010 (which is repeated there twice).

Response:

We appreciate that you have noticed this problem. We have corrected the presentation. In addition, we have carefully checked the whole text to avoid similar problems. The corrections are shown below. You can see it in P10 L331-332.

Original expression: “In 2006 and 2010, the density of LCZ 1 (compact high-rise) was the lowest. In 2010 and 2020, LCZ 10 (heavy industry) was the lowest.”

Revised expression: “In 2006 and 2010, the proportion of LCZ 1 (compact high-rise) is the smallest. In 2016 and 2020, the proportion of LCZ 10 (heavy industry) is the smallest.”

L242-L243: What is the exact meaning of the sentence: LCZ 8 was the LCZ with the biggest increase (focusing on Changsha) or LCZ 8 grew the most in Changsha (compared to other cities) - the overall context of the results leads me to a different conclusion than the sentence implies grammatically. This is not the only strange sentence, but rather one of many examples and someone to smooth the text into a more reader-friendly version would be much needed.

Response:

Thanks for your comments. We have modified the expression of this part. The problem to be explained here is that LCZ8 has the largest area increase, but the area change rate is not the largest. In this paper, LCZ1 has the largest area change rate. This shows that the increase area of LCZ1 is not more than that of LCZ8, but the construction speed of LCZ1 in these years is far more than LCZ8. You can see in P12 L348-353.

Original expression: “From 2006 to 2020, the LCZ 8 (large low-rise) area increased the most in Changsha. It shows a significant increase. The second largest increase is in LCZ 1 (compact high-rise). LCZ 9 (sparsely built) has the largest reduction in area, followed by LCZ 2 (compact mid-rise). LCZ 8 has the largest area increase in Zhuzhou, followed by LCZ 1. The largest area reduction is in LCZ 9, followed by LCZ 5 (open mid-rise). Xiangtan is the same as Zhuzhou.”

Revised expression: “From 2006 to 2020, the total area of LCZ 8 (large low-rise) has increased the most, followed by LCZ 1 (compact high-rise) in the CZT urban agglomeration. In Changsha, LCZ 9 (sparsely built) has the largest area reduction, followed by LCZ 2 (compact mid-rise). In Zhuzhou and Xiangtan, LCZ 9 has the largest reduction in area, followed by LCZ 5 (open mid-rise). The area of LCZ A (dense trees) and LCZ G (water) in the CZT urban agglomeration has largely remained unchanged.”

L242: It shows a significant increase “ – why is the meaning of the previous sentence so strangely repeated - was it not enough to connect the information about significance with the preceding sentence?

Response:

We appreciate that you have noticed this problem. The meaning of this sentence is repeated with the previous one, and we deleted this sentence. You can see in P12 L354.

L254-L256: These sentences don't make much sense without the following sentences. However, I think that it would be better to connect the text organically so that it makes sense to the reader directly.

Response:

Thanks for your suggestion. We have reformulated the meaning of this text, which means the transfer of the LCZ area. You can see it in detail in P14 L369-371.

From 2006 to 2020, in the CZT urban agglomeration, the transfer between LCZ 2/LCZ 1, LCZ 5/LCZ 2, LCZ 9/LCZ 6 (open low rise), LCZ 9/LCZ 8, and LCZ 9/LCZ D (low plants) are close. Between them, LCZ 9 has the largest area converted into LCZ 8.

L276-L278. Study variables – are they explanatory variables or response variables? Which of the named is which. Why is UHII named twice? Is it because it appears as a variable in two different models (One with LCZ and the other with climate)? Why are these models not specified? Why is it stated in the results that there are some variables involved? This should be in the methodology. Why isn't it just written here if there was a significant effect between LCZ and UHII? And what does the sentence (without verbs) "The LCZ area change rate and city GDP growth rate" even mean?

Response:

Thank you very much for pointing out this issue, and it can make the content of the paper more convincing. We have identified the studied model in the Method section and explained in detail what these variables mean and how they are calculated. We change the expression of the variable “urban heat island intensity (UHII)” to “LST difference” and “climate” to “weather temperature”.

The result we got is that LST of each LCZ has its obvious difference, which can be seen in P16 L393. We have revised the sentence "The LCZ area change rate and city GDP growth rate". This sentence originally intended to express that we studied the correlation between these two variables.

The following is what we added to the model-related variables, which can be seen in P9 L296-315:

In this paper, we took the LCZ area data, the LST data, and the GDP data (Table 34) of the three cities as objects, and used the analysis tools in SPSS software to analyze the correlation of each factor. We build three models. The first variable of each model is the explanatory variable, and the second is the response variable: Model 1, the LCZ area and LST; Model 2, the weather temperature and the LST difference. The weather temperature refers to the average LST values of the CZT urban agglomeration in 2006, 2010, 2016, and 2020. The LST difference refers to the difference in the average LST between LCZs and a city. We use the LST difference to reveal the warming and cooling effects of LCZs; Model 3, the LCZ area change rate, and the GDP change rate. The calculation method for the LCZ area change rate is to divide the change in the LCZ area by the initial LCZ area. For example, the difference obtained by subtracting the LCZ 1 area in 2006 from that in 2010 is divided by the total area of LCZ 1 in 2006, and this value is the LCZ 1 area change rate from 2006 to 2010. The GDP includes the gross production values of the primary industry, secondary industry, and tertiary industry. The calculation method for the GDP is to divide the change in GDP by the initial GDP. The Pearson Correlation Analysis model is applied. The Pearson correlation coefficient r is widely applied in building science research to assess the strength of a two-variable linear association [23,55-57], and its value ranges from -1 to 1. When |r| is closer to 1, it means there is a greater correlation between the two variables. The p-value is used to test the significance of the correlation. When p<0.05, there is a significant correlation, and when p<0.01 there is a very significant correlation.

L278-L279: “the standard regression coefficients, which reflect the correlation of two variables”: During the analysis, the authors have to decide whether there is a correlation between the two variables or a regression. I see no point in mixing these two analyses to analyze one set of variables (and write about them in the same sentence).

Response:

Thank you so much for your suggestion. The analysis model we use is Pearson correlation analysis, so Pearson correlation coefficient is used to represent the correlation between two variables. We have revised the expression of this sentence. You can see in P18 L418:

Original expression: “As shown in Table 5, the values in the color area refer to the standard regression coefficients, which reflect the correlation of two variables.”

Revised expression: “Table 7 shows the Pearson correlation coefficients among the variables.”

L294-L306: There is no such thing as very significant - the results are either significant or not significant, that's the point of a p-value. If the authors want to present proof to the readers that the p-value was really small, or relatively close to the 0.05 limit, then the entire manuscript lacks explicitly written p-values (and test values, degrees of freedom,...)

Response:

Thanks a lot for pointing this out to us. We have added all p-values to Table 7. Pearson correlation among variables and explained them accordingly, you can see in P19 L452-454.

L354: Table 5. Why is this table (along with some graphs) even part of the discussion? Given that individual LCZs are correlated to some extent, it would be appropriate to adjust all p.values obtained by tests using adjustment (e.g. false discovery rate).

Response:

Thanks for your suggestion. It helps our paper structure optimization and reading. We have adjusted the position of the diagrams in the text and linked them with the corresponding text.

L361: “High-rise buildings (LCZ 1, 4) and open low-rise buildings (LCZ 6) are more capable of dissipating heat than they are”. This sentence doesn't make sense unless the authors meant something else by “they” - for example, the factors listed a few sentences back. In that case, instead of “they”, it was necessary to explicitly write "above-mentioned factors" or something similar.

Response:

Thank you very much for your review. This helps our paper language to be clearer. We have revised the expression of this sentence. The revision is shown below. You can see in P20 L493-497.

2020

Original expression: “This may be due to their low air circulation and weak heat dissipation capacity. High-rise buildings (LCZ 1, 4) and open low-rise buildings (LCZ 6) are more capable of dissipating heat than they are.”

Revised expression: “This may be due to the low air circulation and weak heat dissipation capacity of the compact mid-rise, the compact low-rise, and the open mid-rise. The cooling capacity of the high-rise buildings and the open low-rise is stronger than that of the compact mid-rise and low-rise, and the open low-rise.”

L405: Why do LCZs (if several are listed consecutively) cease to be listed by their full name from this line and, inconsistently with the rest of the manuscript, are listed only by the final letter/number?

Response:

Thanks for pointing out the problem. It is very necessary to unify the abbreviation in the text. For readability, we no longer use abbreviations here. Elsewhere in the text we have unified abbreviations, changing the G to the full expression LCZ G. You can see in P21 L547-549.

Original expression: “The UHII of LCZ 2 (compact mid-rise), 3, 5, D, and G is significantly affected by climate change.”

Revised expression: “The LST differences of the compact mid-rise, the compact low-rise, the open mid-rise, the low plants, and the water zones are significantly affected by the change in weather temperature.”

L411: ineffective rather than ineffectively? 

Response:

Thank you for your suggestion. We have changed from “ineffectively” to “ineffective”, and double-checked the use of the word in the paper. You can see in P21 L557

Revised expression: “Therefore, the compact low-rise is ineffective against climate warming and inhibits economic growth.”

Reviewer 2 Report

General comment:

Don’t use the term climate throughout the manuscript, you don’t use climatic data, only the retrieved LST.

Be more precise on the GDP, and provide number of the growth for each LCZ.

Author Response

Dear Reviewer,

We are grateful for your time and effort on reviewing the manuscript. The new manuscript has been revised based on your comments and the corrections made are explained below.

Hopefully, this revised version would be received favorably.

We are looking forward to hearing from you.

Sincerely yours,

Authors

☆ ☆ ☆ ☆ ☆

-Reviewer 2-

General comment:

Don’t use the term climate throughout the manuscript, you don’t use climatic data, only the retrieved LST.

Be more precise on the GDP, and provide number of the growth for each LCZ.

Section 1

Add an explanatory phrase of the coping and pulling effects.

Response:

Thank you for your suggestion. We explained “coping effect” in 2.

We changed “pulling effects” to “economic benefits”, which have been explained in 3. You can see in details in P3 L114-121.

Original expression: “Therefore, this paper takes the LCZ system as the framework to study the coping effects of different land covers on climate warming. We explore the pulling effect of different LCZs on urban gross domestic product (GDP).”

Revised expression: “In view of this, we propose an improved WUDAPT method and take the LCZ system as the framework to analyze: 1. the correlation between LST and the total areas of LCZ; 2. under a high weather temperature, which LCZ types will be affected the most; and 3. which industries’ GDP will be impacted the most by LCZ changes. The purpose of this is to judge: 1. Whether the change in the total area of an LCZ type will affect the aver-age LST; 2. How the LST in different LCZ types will vary according to weather temperature changes; and 3. The economic benefits of each LCZ. Finally, we provide relevant recommendations for urban construction.”

Lines 40-43: Move these lines at the last paragraph and rephrase it, clearly defining there the goal of this paper.

Response:

Thank you for your suggestion. We have restructured these sentences into the last paragraph of “1. Introduction” and have stated the research content and objectives of this paper in more detail. For details, you can see it in P3 L112-121.

At present, there are relatively few studies on the spatial heterogeneity of UHI changes along with weather temperatures. Moreover, we need to efficiently generate LCZ maps for multiple years in multiple adjacent cities. In view of this, we propose an improved WUDAPT method and take the LCZ system as the framework to analyze: 1. the correlation between LST and the total areas of LCZ; 2. under a high weather temperature, which LCZ types will be affected the most; and 3. which industries’ GDP will be impacted the most by LCZ changes. The purpose of this is to judge: 1. Whether the change in the total area of an LCZ type will affect the average LST; 2. How the LST in different LCZ types will vary according to weather temperature changes; and 3. The economic benefits of each LCZ. Finally, we provide relevant recommendations for urban construction.

Section 3

Lines 157-162: Describe more in depth the time series sample transfer method.

Response:

Thank you very much for pointing out the issue for us. We have described the time-series sample transfer method in more detail. You can see it in P7 L211-221:

First, we used change vector analysis to assess whether a pixel has changed between 2020 and 2016 using the multispectral and thermal infrared bands of the Landsat im-ages. We calculated the gray value of a pixel in 2020 and 2016, and then calculated the change vector and the change magnitude. When the change magnitude exceeds a certain threshold, the pixel is identified as a changed pixel. Second, the reliability of the pixels is checked by the specificity of a probability distribution. Finally, based on in-variability and reliability, the k-nearest neighbor algorithm is used to further select representative training areas. To obtain the CZT urban agglomeration training areas for 2016, we deleted the training areas whose pixels changed from 2020 to 2016 and added new training areas. The training areas of 2010 and 2006 were obtained by the same method [23,53].

Lines 173-177: Provide a table with the statistical metrics of the validation of LCZ.

Response:

Thank you for your suggestion. We have added a process for LCZ map accuracy verification. We use the accuracy verification report sent by WUDAPT to judge the LCZ map accuracy. You can see it in P7 L237-245,585. We have added Figure A1 (P23) in the Appendix.

P7 L237-245: To ensure the quality of the generated LCZ maps, the WUDAPT applies an automatic cross-validation approach using 25 bootstraps. In each bootstrap, 70% of the training area polygons are used for training, and 30% are used for testing. The polygons are chosen by stratified (LCZ-type) random sampling, while the original LCZ class frequency distribution is kept. This process is repeated 25 times to provide confidence intervals for the accuracy measures. In addition, this approach also allows the creation of a probability map that represents how often (in %) a pattern was mapped in the iterative procedure. The resulting LCZ maps are based on all training areas, and the overall accuracy indicates the percentage of correctly classified pixels.

Lines 194-197: Provide a table with the statistical metrics of the validation of LST.

Response:

Thank you for your suggestion. The supplement of the LST data validation is in P8 L266-271, and we added table3:

The reliability of the LST results is verified by comparing the average temperature be-tween the recorded temperature data and the LST values retrieved from remote sensing. The average difference between the two sets of data is found to be less than 1°C (Table 3).

Lines 208-212: Jusstify the chosen 1 standard devation for UHI through references to other works and/or through arguments.

Response:

Thank you so much for pointing this out. We no longer use UHII in this paper but directly use LST. We use the standard deviation method to divide the LST values into seven levels, each of which differs by one standard deviation. In this paper we no longer define the urban heat island and the urban cold island. Specifically, you can see it in P9 L282-286.

First, we calculate the average LST of the CZT urban agglomeration. We then use the standard deviation method to classify the LST values into seven categories: extremely high temperature, high temperature, relatively high temperature, medium temperature, relatively low temperature, low temperature, and extremely low temperature [22].

Line 222: Figure 3 doesn’t exist, add it.

Response:

Thanks for your correction. We have supplemented Figure 3, which you can see in P11 L342.

Section 4

Lines 274-275 and 277-278: Remove them, they are irrelevant in the section.

Response:

Thank you for your suggestion. We have moved the variables into the Method section and added descriptions and calculations of the variables. For details, you can see it in P9 L298-310

Lines 278-307: You refer to Table 5, but you talk about rendering and colors of a figure. Needs major revision. Clearly define where you refer to, which Figure etc. I don’t understand where you refer in these paragraphs.

Response:

Thanks for pointing out the problem. We have revised our statement. Here is to illustrate Figure 7. The darker the color, the larger the difference between the LST of the LCZ and the average LST. For details, you can see it in P16 L389-392.

Figure 7 shows the LST difference of the CZT urban agglomeration. The darker the color, the greater the LST difference between LCZs and the city. which means that the stronger the warming or cooling effect. The "*" indicates a high or low temperature, and the "**" indicates an extremely high or extremely low temperature.

Section 5

Lines 323 and 329: “Two type” cities/industries? Please clarify.

Response:

Thanks for your question. A two-type city refers to a natural resource-saving and environment-friendly city, and a two-type industry refers to a natural resource-saving and environment-friendly industry. Two-type is mentioned in the Study area section, you can see it in P3 L134-135.

The aim is to build them into "two-type" cities (national resource-saving type and national environmentally friendly type) [49].

Lines 343-344: Complete the sentence.

Response:

Thanks for pointing out the problem. This is due to a typography problem that separates the text from the picture. We have adjusted the position of the chart and text and placed them in the corresponding position for easy reading, thank you!

Reviewer 3 Report

This paper introduces an interesting approach to measure urban heat island effects across three example cities. The topic lands on an impactful area, and I believe will be of interest to the readers. I also appreciated the effort made in the analysis by the authors. Most of the paper is generally well-written, but I would recommend some large restructuring, particularly around parts in the method, results and discussion. Some additional references have also been suggested. Please refer to the detailed comments below for more information. I would also recommend explaining key parts of the paper in more detail. Some variables and the verification stage were difficult to understand. I hope that the authors find the following comments useful:

#1: P1, L38: “human activity” could probably be stated as “anthropomorphic activity.” Some examples could also be given here, such as air-conditioning, pollution levels, etc.

#2: P1, L40-44: This sentence appeared quite abruptly here, and could be moved later in this section. The authors have not yet mentioned thermal comfort, the human living environment, or economic effects of urban land, so readers would not yet understand their relevance to the optimization of urban construction layout.

#3: P2: I though the authors could have briefly mentioned the influence of surface albedo and mitigation approaches in dense urban spaces (e.g. green roofs.) There is a lot of literature in area (e.g., Santamouris, 2014. Cooling the cities–a review of reflective and green roof mitigation technologies to fight heat island and improve comfort in urban environments; Santamouris et al. 2011. Using advanced cool materials in the urban built environment to mitigate heat islands and improve thermal comfort conditions), but was not mentioned in the passages that related to this. It would be useful to expand on other efforts made toward reducing urban heat island effects.

#4: The references [6, 21] for the Pearson’s correlation coefficient could have been improved; these could have included statistical papers. For example, please refer to: Pearson, 1805. Notes on regression and inheritance in the case of two parents.

#5: Section 2 could be labeled as the “Method” with “Study Area” as 2.1, and “Materials and Methods” as 2.2. Please also restructure the following sub-sections accordingly. For the study area, it would have been helpful to also include the average minimum and maximum temperatures for the Hunan Province, and if available, the temperatures inside the city center compared to the average, highlighting the urban heat island effect for this area.

#6: P4, L136: Please clarify what is the horizontal scale, and from what point does this measure begin.

#7: The authors could expand on what it meant by accuracy in the context of their work, and provide more detail for how this was actually verified. This part was not particularly clear.

#8: 3.4: The authors specified that the correlation coefficient is used extensively in academic research, but only one reference is provided. Perhaps this could be restated to mention that it is widely applied in building science research to be more precise, and to provide more recent examples of its use: For example, Liu et al 2015. Correlation analysis of building plane and energy consumption of high-rise office building in cold zone of China; Kent et al 2021. A data-driven analysis of occupant workspace dissatisfaction; Rauf et al 2020. Analysis of correlation between urban heat islands (UHI) with land-use using sentinel 2 time-series image in Makassar city.

The authors may also want to apply benchmarks to help denote the strength of the associations that they were measuring between different variables (please refer to: Kent et al 2021.) This would have been very helpful when interpreting Table 5 on page 14.

#9: Shouldn’t Figure 3 belong in the results section near Section 4.1.1? Unless I missed it, I was also not able to find the proportions the authors were refer to in this section. I think that the results may be referring to those found in Figure 4, showing the LCZ areas (not proportions) for each city. If this is correct, I would recommend the results are more clearly structured in an improvement order of presentation to avoid confusing readers.

#10: Section 4.1.2: “Figure 4 (a) explains” please change to “Figure 4 (a) shows.” Instead of narrating the changes, for highest to lowest, for each city, it may have been simpler for readers to see these changes by calculating the differences and then ranking them, and placing this information above the plots in Figure 4.

#11: Section 4.2: Please consider restructuring this part of the paper. Figures 13 and 14, and Table 4 are found much further in paper in another section, making it unclear which part of the paper the authors were referring to. I would also strongly urge the authors to move the figures and table into the results section, and reorganize the order of the subsections. This would allow the figures and tables to appear much closer to the relevant text that describes them.

#12: Figure 4: Please explain the numbers that appear inside the plots. The values were always larger than the y-axis scale, and I wasn’t able to grasp there relevance to the plots.

#13: Figure 6: Would it have made more sense to create the same scale (e.g. LST 9oC to 33oC) for ever spatiotemporal distribution map shown in the figure? This would have made each more comparable, without the need to plotting the same legend for each map. The same could be applied to Figure 7, since the legend makes it very difficult to compare the plots for 2006 and 2016 to 2010 and 2020.

#14: Table 5: The variables listed on the first column could be explained in more detail. The authors have not yet explained each, and in particularly, it was not clear what “Climate” was or how it is measured. Please also consider applying benchmarks to the correlation coefficients, so that the size of the associations can be compared in more detail.

Author Response

Dear Reviewer,

We are grateful for your time and effort on reviewing the manuscript. The new manuscript has been revised based on your comments and the corrections made are explained below.

Hopefully, this revised version would be received favorably.

We are looking forward to hearing from you.

Sincerely yours,

Authors

☆ ☆ ☆ ☆ ☆

-Reviewer 3-

Responses to comments from the reviewers:

This paper introduces an interesting approach to measure urban heat island effects across three example cities. The topic lands on an impactful area, and I believe will be of interest to the readers. I also appreciated the effort made in the analysis by the authors. Most of the paper is generally well-written, but I would recommend some large restructuring, particularly around parts in the method, results and discussion. Some additional references have also been suggested. Please refer to the detailed comments below for more information. I would also recommend explaining key parts of the paper in more detail. Some variables and the verification stage were difficult to understand. I hope that the authors find the following comments useful:

#1: P1, L38: “human activity” could probably be stated as “anthropomorphic activity.” Some examples could also be given here, such as air-conditioning, pollution levels, etc.

Response:

Thank you for your suggestion. We have changed “human activity” to “anthropomorphic activities”. Then we add some examples. You can see in the text P1 L41-42:

"UHI is directly linked to anthropomorphic activities, such as the direct heat emissions of fuel combustion and electricity consumption "

#2: P1, L40-44: This sentence appeared quite abruptly here, and could be moved later in this section. The authors have not yet mentioned thermal comfort, the human living environment, or economic effects of urban land, so readers would not yet understand their relevance to the optimization of urban construction layout.

Response:

Thank you very much for pointing out the problem, which helps the logic of our paper to be smooth. We move the purpose to P3 L118-121 and modified the expression to connect the context. Specifically, you can see in the text:

The purpose of this is to judge: 1. Whether the change in the total area of an LCZ type will affect the average LST; 2. How the LST in different LCZ types will vary according to weather temperature changes; and 3. The economic benefits of each LCZ. Finally, we provide relevant recommendations for urban construction.

We have added background content on the relationship between urban heat island, Local climate zone (LCZ) change, and land economic benefits, which you can see in P1-2 L44-52:

UHI affects people's lives and the environment in many ways, which not only affects the quality of human life but also affects the local climate, seriously hindering the sustainable development of the urban environment [6-8]. The deterioration of the environment will limit the social and economic benefits of land use [9]. Local climate zone (LCZ) classifies the land surface according to different building densities, geo-metric shapes, and surface characteristics, which can describe the land development intuitively. Therefore, it is helpful for the rational development of urban land to clarify the relationship between UHI, LCZ changes, and their economic implications to explore the influencing factors.

#3: P2: I though the authors could have briefly mentioned the influence of surface albedo and mitigation approaches in dense urban spaces (e.g. green roofs.) There is a lot of literature in area (e.g., Santamouris, 2014. Cooling the cities–a review of reflective and green roof mitigation technologies to fight heat island and improve comfort in urban environments; Santamouris et al. 2011. Using advanced cool materials in the urban built environment to mitigate heat islands and improve thermal comfort conditions), but was not mentioned in the passages that related to this. It would be useful to expand on other efforts made toward reducing urban heat island effects.

Response:

Thank you so much for your suggestion. Complementary approaches to urban heat island mitigation are highly desirable. We have added to the Introduction the impact of surface albedo on urban heat islands, and the mitigation of urban heat islands by green roof reflective roofs, and the references you mentioned to the text. Specifically, you can see in the text P2 L69, 72-73,76-80.

For instance, a high-albedo roof can help reduce ambient temperature [29,30].

The studies propose that in order to alleviate UHI, urban planning should rationally arrange parks, green spaces, and water ponds; avoid high-density low-rise residential areas with low vegetation coverage; increase inner-city surface roughness; and use reflective roofs, green roofs, and vertical greening systems [29-33].

#4: The references [6, 21] for the Pearson’s correlation coefficient could have been improved; these could have included statistical papers. For example, please refer to: Pearson, 1805. Notes on regression and inheritance in the case of two parents.

Response:

Thanks for the suggestion. Using more influential references is much needed. So, we read the reference you mentioned, and we cited it, you can see in P2 L94.

#5: Section 2 could be labeled as the “Method” with “Study Area” as 2.1, and “Materials and Methods” as 2.2. Please also restructure the following sub-sections accordingly. For the study area, it would have been helpful to also include the average minimum and maximum temperatures for the Hunan Province, and if available, the temperatures inside the city center compared to the average, highlighting the urban heat island effect for this area.

Response:

Thank you very much for pointing out the problem. We have restructured the paragraph. In the study area section, we supplemented the urban heat island effect in Changsha, Zhuzhou, and Xiangtan. You can see the specific content in the text P3 L136-151:

The CZT urban agglomeration has a subtropical monsoon climate with hotter thermal conditions than other cities in Hunan Province. In China, Changsha is one of the four hottest cities during summer. The high temperatures in Hunan Province are concentrated in July and August. In the past five years, the average July temperature in Hu-nan Province has risen from 29.2°C to 29.5°C, an increase of 0.3°C. In Changsha, Zhuzhou, and Xiangtan, the average temperature in July increased by 0.1°C, 1.0°C, and 0.6°C, respectively. Especially in 2022, there are 85 days when the average daily temperature in summer is above 30°C and approximately 30 days when the temperature is higher than 35°C. This shows a serious urban heat island problem [50-52]. The UHI of the CTZ urban agglomeration is mainly distributed in the built-up area, with growing intensity. In the past two decades, the UHI in the central urban area of the CZT urban agglomeration has gradually increased from 3.3°C to 10.1°C.

#6: P4, L136: Please clarify what is the horizontal scale, and from what point does this measure begin.

Response:

Thanks for your question. To determine the LCZ type, remote sensing maps are typically used in conjunction with 3D street view maps and the field investigation method The horizontal scale refers to the level in the two-dimensional direction of the remote sensing image. To avoid misunderstandings, we have removed the term “the horizontal scale”. Specifically, you can see it in P5 L185

#7: The authors could expand on what it meant by accuracy in the context of their work, and provide more detail for how this was actually verified. This part was not particularly clear.

Response:

Thanks for asking this question. We have supplemented the process of LCZ map accuracy verification, which you can see in the paper P7 L237-245, and We have added Figure A1. Accuracy reports sent by WUDAPT (P23 L584) in the Appendix:

P7 L237-245: To ensure the quality of the generated LCZ maps, the WUDAPT applies an automatic cross-validation approach using 25 bootstraps. In each bootstrap, 70% of the training area polygons are used for training, and 30% are used for testing. The polygons are chosen by stratified (LCZ-type) random sampling, while the original LCZ class frequency distribution is kept. This process is repeated 25 times to provide confidence intervals for the accuracy measures. In addition, this approach also allows the creation of a probability map that represents how often (in %) a pattern was mapped in the iterative procedure. The resulting LCZ maps are based on all training areas, and the overall accuracy indicates the percentage of correctly classified pixels.

The supplement of LST data validation is in P8 L266-271, and we added Table3:

The reliability of the LST results is verified by comparing the average temperature be-tween the recorded temperature data and the LST values retrieved from remote sensing. The average difference between the two sets of data is found to be less than 1°C (Table 3).

#8: 3.4: The authors specified that the correlation coefficient is used extensively in academic research, but only one reference is provided. Perhaps this could be restated to mention that it is widely applied in building science research to be more precise, and to provide more recent examples of its use: For example, Liu et al 2015. Correlation analysis of building plane and energy consumption of high-rise office building in cold zone of China; Kent et al 2021. A data-driven analysis of occupant workspace dissatisfaction; Rauf et al 2020. Analysis of correlation between urban heat islands (UHI) with land-use using sentinel 2 time-series image in Makassar city.

The authors may also want to apply benchmarks to help denote the strength of the associations that they were measuring between different variables (please refer to: Kent et al 2021.) This would have been very helpful when interpreting Table 5 on page 14.

Response:

Thank you very much for pointing out the problem. We have modified the expression here and provided an up-to-date example. We have included p-values in Table 7 to illustrate the strength of the correlation. Specifically, you can see it in P9 L311-315, P19 452-454:

The Pearson Correlation Analysis model is applied. The Pearson correlation coefficient r is widely applied in building science research to assess the strength of a two-variable linear association [23,55-57], and its value ranges from -1 to 1. When |r| is closer to 1, it means there is a greater correlation between the two variables. The p-value is used to test the significance of the correlation. When p<0.05, there is a significant correlation, and when p<0.01 there is a very significant correlation.

#9: Shouldn’t Figure 3 belong in the results section near Section 4.1.1? Unless I missed it, I was also not able to find the proportions the authors were refer to in this section. I think that the results may be referring to those found in Figure 4, showing the LCZ areas (not proportions) for each city. If this is correct, I would recommend the results are more clearly structured in an improvement order of presentation to avoid confusing readers.

Response:

Thank you very much for pointing out the problem. It is very helpful for us. We added Table 5 to the text, indicating the proportion of LCZs, and put Figure 3 in the corresponding position of the text. We have adjusted all the charts and text positions to make it easier to read. You can see Table 5 in P12 L343.

#10: Section 4.1.2: “Figure 4 (a) explains” please change to “Figure 4 (a) shows.” Instead of narrating the changes, for highest to lowest, for each city, it may have been simpler for readers to see these changes by calculating the differences and then ranking them, and placing this information above the plots in Figure 4.

Response:

Thank you for your question. We have made corresponding revisions and revised other expressions with this kind of problem in the paper. We modified Figure 4(a) by calculating the area change of LCZs so that it is easier to see the change in area and compare the differences for each city. Specifically, you can see in P12-13 L348-353,368:

From 2006 to 2020, the total area of LCZ 8 (large low-rise) has increased the most, followed by LCZ 1 (compact high-rise) in the CZT urban agglomeration. In Changsha, LCZ 9 (sparsely built) has the largest area reduction, followed by LCZ 2 (compact mid-rise). In Zhuzhou and Xiangtan, LCZ 9 has the largest reduction in area, followed by LCZ 5 (open mid-rise). The area of LCZ A (dense trees) and LCZ G (water) in the CZT urban agglomeration has largely remained unchanged.

#11: Section 4.2: Please consider restructuring this part of the paper. Figures 13 and 14, and Table 4 are found much further in paper in another section, making it unclear which part of the paper the authors were referring to. I would also strongly urge the authors to move the figures and table into the results section, and reorganize the order of the subsections. This would allow the figures and tables to appear much closer to the relevant text that describes them.

Response:

Thank you for your suggestion. We have adjusted the positions of figures and text in the text and reorganized the order of the summaries for easier reading.

#12: Figure 4: Please explain the numbers that appear inside the plots. The values were always larger than the y-axis scale, and I wasn’t able to grasp there relevance to the plots.

Response:

Thank you for pointing out the problem. Figure 4(b) expresses the rate of change of the LCZ area, which is to evaluate the rate of change of the LCZ. We have supplemented the calculation of the rate of change of the LCZ area in detail. Some values exceed the scale of the y-axis, because the rate of change is particularly large. For the aesthetics of the drawing, the corresponding values are marked on the excess part. This mapping method is also used in some papers (e.g. Lu, Y.; Yang, J.; Ma, S. Dynamic changes of local climate zones in the Guangdong–Hong Kong–Macao Greater Bay Area and their spatio-temporal impacts on the surface urban heat island effect between 2005 and 2015.). You can see the specific content in P9 L303-308:

Model 3, the LCZ area change rate, and the GDP change rate. The calculation method for the LCZ area change rate is to divide the change in LCZ area by the initial LCZ area. For example, the difference obtained by subtracting the LCZ 1 area in 2006 from that in 2010 is divided by the total area of LCZ 1 in 2006, and this value is the LCZ 1 area change rate from 2006 to 2010.

#13: Figure 6: Would it have made more sense to create the same scale (e.g. LST 9oC to 33oC) for ever spatiotemporal distribution map shown in the figure? This would have made each more comparable, without the need to plotting the same legend for each map. The same could be applied to Figure 7, since the legend makes it very difficult to compare the plots for 2006 and 2016 to 2010 and 2020.

Response:

Thanks a lot for your suggestion. It's very helpful. We have unified the legend for Figure 6 and Figure 7. You can see in P15 L382, P17 L410.

#14: Table 5: The variables listed on the first column could be explained in more detail. The authors have not yet explained each, and in particularly, it was not clear what “Climate” was or how it is measured. Please also consider applying benchmarks to the correlation coefficients, so that the size of the associations can be compared in more detail.

Response:

Thank you very much for pointing out this issue, and it can make the content of the paper more convincing. We have identified the studied model in the Method section and explained in detail what these variables mean and how they are calculated. We change the expression of the variable “urban heat island intensity (UHII)” to “LST difference” and “climate” to “weather temperature”. We added the p-values in table 7

, which can be seen in P9 L296-315, P19 L452.

In this paper, we took the LCZ area data, the LST data, and the GDP data (Table 34) of the three cities as objects, and used the analysis tools in SPSS software to analyze the correlation of each factor. We build three models. The first variable of each model is the explanatory variable, and the second is the response variable: Model 1, the LCZ area and LST; Model 2, the weather temperature and the LST difference. The weather temperature refers to the average LST values of the CZT urban agglomeration in 2006, 2010, 2016, and 2020. The LST difference refers to the difference in the average LST between LCZs and a city. We use the LST difference to reveal the warming and cooling effects of LCZs; Model 3, the LCZ area change rate, and the GDP change rate. The calculation method for the LCZ area change rate is to divide the change in the LCZ area by the initial LCZ area. For example, the difference obtained by subtracting the LCZ 1 area in 2006 from that in 2010 is divided by the total area of LCZ 1 in 2006, and this value is the LCZ 1 area change rate from 2006 to 2010. The GDP includes the gross production values of the primary industry, secondary industry, and tertiary industry. The calculation method for the GDP is to divide the change in GDP by the initial GDP. The Pearson Correlation Analysis model is applied.

Round 2

Reviewer 1 Report

I appreciate that the authors have responded to each of my comments in great detail and modestly. Nevertheless, I still find several things in the manuscript that I propose to improve.

In the results, it is often written what the reader will see in the figures and this information is or is not presented also with the figure - Instead, it would be better if the results focus on the verbal description of the results and only refer to the figures, and figures are described separately. The best example is probably figure 7, which is explained on lines L323-L325, then the same explanation is repeated directly below the figure on lines L340-L342, with both the caption (but only partially!) and the figure repeated on lines L338-L340.

Table 7. Please provide the p-value rounded to three decimal places. I also still think that due to the multiple comparison of interdependent factors (individual types of LCZ), it would be appropriate to perform an adjustment (e.g. FDR) for the P-value, which would reduce the number of significant results.

Author Response

Dear Reviewer,

We are grateful for your time and effort on reviewing the manuscript. The new manuscript has been revised based on your comments and the corrections made are explained below.

Hopefully, this revised version would be received favorably.

We are looking forward to hearing from you.

Sincerely yours,

Authors

☆ ☆ ☆ ☆ ☆

-Reviewer 1-

I appreciate that the authors have responded to each of my comments in great detail and modestly. Nevertheless, I still find several things in the manuscript that I propose to improve.

Responses to comments from the reviewer:

In the results, it is often written what the reader will see in the figures and this information is or is not presented also with the figure - Instead, it would be better if the results focus on the verbal description of the results and only refer to the figures, and figures are described separately. The best example is probably figure 7, which is explained on lines L323-L325, then the same explanation is repeated directly below the figure on lines L340-L342, with both the caption (but only partially!) and the figure repeated on lines L338-L340.

Response:

Thank you very much for your comments, which help to improve the quality of the paper. In the Results section, we state the results and only refer to the figures, and we describe the figure in the title of this figure. You can see in P13 L327, P14 L355-357, and P15, L362-364. Following are instances of modifications:

Original expression: “Figure 6 shows the spatial distribution of LST. Table 4 shows the average LST of each LCZ”

Revised expression: “As shown in Figure 6, the extremely high and moderately high temperature areas of the CZT urban agglomeration are mainly distributed along the Xiangjiang River in 2006”

Revised expression: “The extremely high and moderately high temperatures are distributed in LCZ 8 and LCZ 10 (Figure 7).”

Revised expression: “Figure 7. Land surface temperature (LST) differences between LCZs and the cities of Changsha, Zhuzhou, and Xiangtan in 2006, 2010, 2016, and 2020. The "*" indicates a moderately high or moderately low temperature. the "**" indicates an extremely high or extremely low temperature.”

In order to describe the figures clearly, we supplemented the landmark place names of Changsha, Zhuzhou, and Xiangtan in Figure 1, Figure 3, and Figure 6, and we reorganized the results of the figures, in P9-10 L284-293, P13 L331-344.

Table 7. Please provide the p-value rounded to three decimal places. I also still think that due to the multiple comparison of interdependent factors (individual types of LCZ), it would be appropriate to perform an adjustment (e.g. FDR) for the P-value, which would reduce the number of significant results.

Response:

We appreciate your proposal, which serves to strengthen the persuasiveness of our results. First, we rounded both Pearson correlation coefficients and p-values to three decimal places. Second, we adjusted the p-values using the FDR method, in Table 7, in P16 L386. According to the adjusted p-values, we reduced the significant results between LCZ 4 (open high-rise) with the primary industry, and between LCZ G (water) with the tertiary industry. You can find details in P17 L439-443.

Reviewer 2 Report

Can be published

Author Response

Dear Reviewer,

We are grateful for your time and effort on reviewing the manuscript.

Many thanks! Have a good day!
Best regards,

Reviewer 3 Report

Thank you for revising your work based on my previous comments. This was highly appreciated. I thought your responses were detailed, and I believed that this had helped improved the quality of this paper. I have some further (minor) comments, which I thought would be useful to consider below:

#1. Page 2, line 66: It would be helpful to elaborate this sentence. For example, a high surface albedo reduces the conversion of daylight into heat. 

#2. Page 3, line 125: Please amend to: 'has risen by 0.3oC from 29.2oC to 29.5oC.'

#3: Section 2.2.1. Perhaps the numbering from 1 to 3 can be changed to: Part 1, Part 2, and Part 3, respectively, so as to not confuse readers with section heading numbers.  Similarly, this can also be applied to section 2.2.2, and 2.2.3.

4: Correlation: It would be useful to also describe what values closer to -1 represent (e.g. an inverse relation between the two variables.) Later in the paper, some negative correlations were shown, but this hadn't been explained.

5: Page 15, line 331: Please check: 'extremely high and high temperatures.' Maybe this should be 'extremely high to moderately high temperatures.' This could also be applied on line 332 for cold temperatures.

Author Response

Dear Reviewer,

We are grateful for your time and effort on reviewing the manuscript. The new manuscript has been revised based on your comments and the corrections made are explained below.

Hopefully, this revised version would be received favorably.

We are looking forward to hearing from you.

Sincerely yours,

Authors

☆ ☆ ☆ ☆ ☆

-Reviewer 3-

Thank you for revising your work based on my previous comments. This was highly appreciated. I thought your responses were detailed, and I believed that this had helped improved the quality of this paper. I have some further (minor) comments, which I thought would be useful to consider below:

Responses to comments from the reviewer:

#1. Page 2, line 66: It would be helpful to elaborate this sentence. For example, a high surface albedo reduces the conversion of daylight into heat.

Response:

Thank you for your suggestion. We have reformulated the sentence. You can see in P2 L67:

Original expression: “For instance, a high-albedo roof can help reduce ambient temperature”

Revised expression: “A high surface albedo reduces the conversion of daylight into heat.”

#2. Page 3, line 125: Please amend to: 'has risen by 0.3oC from 29.2oC to 29.5oC.'

Response:

Thank you for your suggestion. We revised the sentence, in P3 L126-127.

Original expression: “In the past five years, the average July temperature in Hunan Province has risen from 29.2°C to 29.5°C, an increase of 0.3°C.”

Revised expression: “In the past five years, the average July temperature in Hunan Province has risen by 0.3°C from 29.2°C to 29.5°C.”

#3: Section 2.2.1. Perhaps the numbering from 1 to 3 can be changed to: Part 1, Part 2, and Part 3, respectively, so as to not confuse readers with section heading numbers.  Similarly, this can also be applied to section 2.2.2, and 2.2.3.

Response:

Thank you so much for noticing this issue. Although we believe that your suggestion is excellent, we follow the format template offered by Sustainability when expressing the subtitle in this format.

4: Correlation: It would be useful to also describe what values closer to -1 represent (e.g. an inverse relation between the two variables.) Later in the paper, some negative correlations were shown, but this hadn't been explained.

Response:

Thanks for your comment. We added the meaning of the statement that the values are closer to -1 in P9 L270-272.

“When r is closer to 1, it reveals a stronger positive correlation between two variables, and when r is closer to -1, it indicates a stronger negative correlation.”

5: Page 15, line 331: Please check: 'extremely high and high temperatures.' Maybe this should be 'extremely high to moderately high temperatures.' This could also be applied on line 332 for cold temperatures.

Response:

Thanks a lot for your suggestion. We reformulated this sentence in P14 L355-357.

“The extremely high to moderately high temperatures are distributed in LCZ 8 and LCZ 10 (Figure 7). The moderately low to extremely low temperatures are distributed in LCZ A and LCZ G.”

We changed the expression of “high temperature” to “moderately high temperature”, and “low temperature” to “moderately low temperatures”in this paper.
